# Exploring Concept Subspace for Self-explainable Text-Attributed Graph Learning

## Abstract

We introduce Graph Concept Bottleneck (**GCB**) as a new paradigm for self-explainable text-attributed graph learning. **GCB** maps graphs into a subspace—a *concept bottleneck*—where each concept is a meaningful phrase, and predictions are made based on the activation of these concepts. Unlike existing interpretable graph learning methods that primarily rely on subgraphs as explanations, the concept bottleneck provides a new, semantically grounded form of interpretation that is human-readable. To construct this concept space, we contrastively pretrain a graph encoder so that graphs are aligned with a concept embedding space, employ large language models to propose candidate concepts at both the instance and the class level, and apply the information bottleneck principle to retain a compact, causally informative subset for the downstream predictor. This pipeline not only yields more concise and faithful explanations but also explicitly guides the model to "think" toward the correct decision. We empirically validate **GCB** on five text-attributed node classification benchmarks, and find that **GCB** achieves intrinsic interpretability with accuracy on par with black-box Graph Neural Networks (GNNs). Moreover, **GCB**'s advantage over GNNs increases under distribution shifts and adversarial perturbations, suggesting that its robustness arises from concept-guided prediction rather than purely graph-structural reasoning.

## 1 Introduction

Trustworthiness has become a critical concern in deployed machine learning such as graph learning in high-stakes domains. An effective way to build trust in AI is to provide interpretations of the prediction process Kakkad et al. (2023). Intrinsic interpretability, which enables models to explain their predictions directly without relying on post-hoc explanations, becomes a particularly desirable property for graph neural networks (GNNs) Miao et al. (2022). Most self-explainable GNNs (SE-GNNs) Azzolin et al. (2025); Dai & Wang (2025); Liu et al. (2025) focus on extracting the most informative yet compressed causal subgraph, which is assumed to be responsible for the prediction and is used for both decision-making and explanation. However, while such subgraphs are typically smaller and contain less redundant information, they are still graphs—often complex and difficult to interpret. This limitation is particularly relevant for text-attributed graphs (TAGs) Yan et al. (2023b), where nodes or edges are associated with natural-language attributes. TAGs are ubiquitous in practice: social-network nodes may contain posts or comments, while e-commerce product nodes are often paired with textual descriptions. These textual attributes provide rich semantic information and offer an opportunity to explain predictions through human-readable concepts rather than graph structures alone. However, current intrinsically interpretable graph models have not fully leveraged this opportunity, leaving a gap between model explanations and the natural-language concepts humans can readily understand.

We narrow the gap between model predictions and human understanding by introducing an intermediate representation - *concept bottleneck*. Specifically, the input graph is mapped to a concept subspace that captures its activations over a set of semantically meaningful concepts. These activations are then mapped to the label through a few feedforward layers for label prediction. In this way, the concept activations serve a dual purpose: they drive the prediction and simultaneously provide explanations for it. Although the general workflow is straightforward, adapting it to graph prediction tasks is non-trivial and introduces several

challenges: (1) *Concept selection:* It is labor-intensive to define concept sets relevant to the prediction task, and graphs often represent abstract structures (e.g., social networks), making it difficult to define and select meaningful, human-interpretable concepts. (2) *Concept alignment:* It remains unclear how to effectively map the input domain (graphs) to the concept domain (language). Unlike vision-language tasks—where models like CLIP Radford et al. (2021) provide off-the-shelf alignment—graphs exhibit irregular structures and high variability across domains, and no such readily applicable model exists. Consequently, a concept predictor must be carefully designed to ensure faithful explanations.

In light of these challenges, we propose **Graph Concept Bottleneck** (**GCB**) as a new paradigm for interpretable text-attributed graph learning. **GCB** consists of three modules. First, we pre-train a universal graph encoder using self-supervised *contrastive concept–graph pretraining*, which aligns graph to the concept space. The encoder can be applied across different downstream datasets. Next, we initialize a concept set through *LLM-enhanced concept retrieval* without manual annotation. This space is further refined by selecting the most informative concepts via *information-constrained bottleneck optimization*. Finally, we train a predictor over the refined concept space to predict the label.

We conduct extensive experiments on five text-attributed graph datasets in clean and perturbed settings to evaluate **GCB**. We highlight two major contributions of **GCB**:

(1) *Concept-based interpretable graph learning framework.* To the best of our knowledge, **GCB** is among the first self-explainable graph learning frameworks that project graph inputs into a language space during prediction. This demonstrates the potential of actively incorporating natural language as an integral part of the reasoning process, enabling more interpretable and transparent deep learning on graph-structured data.

(2) *Robust baseline for text-attributed node classification.* We show that **GCB** performs competitively with SOTA GNNs in standard settings, and its advantages become more pronounced under distribution shifts and data perturbations, establishing a strong baseline for robust and generalizable graph learning.

## 2   Related Work

**Self-explainable GNNs.** In recent years, there has been growing interest in self-explainable GNNs Miao et al. (2022; 2023); Yu et al. (2021; 2022); Wu et al. (2020; 2022); Dai & Wang (2021); Feng et al. (2022); Azzolin et al. (2025); Dai & Wang (2025); Liu et al. (2025); Peng et al. (2024), where the explainability component is integrated into the prediction process. These methods typically generate informative subgraphs that serve both as explanations and as the basis for predictions. One line of work Miao et al. (2022; 2023); Yu et al. (2021; 2022); Wu et al. (2020) leverages the information bottleneck principle, aiming to extract the most informative yet compact subgraph by optimizing a graph information bottleneck objective. Other approaches Wu et al. (2022); Dai & Wang (2021); Feng et al. (2022) introduce structural constraints to promote interpretability. For example, DIR Wu et al. (2022) decomposes the input graph into causal and non-causal components, enforcing that predictions depend only on the causal part. This line continues to evolve—recent methods extend graph-information-bottleneck self-explanation to temporal graphs Seo et al. (2024), couple rationale extraction with out-of-distribution generalization Yue et al. (2024), and pursue multi-granular structural interpretability Yang et al. (2025). Despite these advances, most existing methods still rely on subgraphs as explanations, whose interpretability is not always guaranteed; indeed, Chen et al. (2024b) shows that such subgraph-importance self-explainers often fail to recover the genuinely relevant structure, motivating explanations grounded in higher-level, human-aligned concepts rather than raw substructure. More recently, researchers have begun exploring alternative forms of explanation. For instance, Bechler-Speicher et al. (2024) proposes Graph Neural Additive Networks, where the relationships between the input graph and the target variable can be directly visualized. Sengupta & Rekik (2025) encodes interpretable cues (e.g., degrees, centrality) into a context vector, which is then mapped to an explanation vector. Müller et al. (2023) employs decision trees to build rule-based predictors that are understandable to humans. However, none of these works employ natural language as a medium for explanations.

**Concept-based explanations for GNNs.** A complementary direction replaces subgraphs with *concepts*—higher-level, human-aligned units of explanation. GCExplainer Magister et al. (2021) clusters last-layer GNN activations into post-hoc concepts, and the Concept Encoder Module Magister et al. (2022) makes such

concepts differentiable and explainable-by-design. Subsequent work extracts global or compositional concepts from trained GNNs through neuron analysis Xuanyuan et al. (2023) or composes local explanations into Boolean logic formulas over learned concepts Azzolin et al. (2023; 2025). A parallel prototype-based family reasons by similarity to learned prototype subgraphs Zhang et al. (2022); Dai & Wang (2025), with PGIB Seo et al. (2023) coupling prototypes to a graph information bottleneck. These concepts are, however, discovered from latent activations and are not natively expressed in natural language, so naming and validating them still requires a human in the loop. **GCB** instead consumes an explicit, natural-language concept space and predicts *through* it, so the same concepts both drive and explain each decision.

**Concept bottleneck models. GCB** builds on Concept Bottleneck Models (CBMs) Koh et al. (2020), which route prediction through human-interpretable concepts. To avoid expensive concept annotation, recent CBMs nominate the concept vocabulary automatically with (vision-)language models: LaBo Yang et al. (2023) and LM4CV Yan et al. (2023a) prompt an LLM for candidate attributes and then select a compact, discriminative subset, while label-free Oikarinen et al. (2023) and post-hoc Yuksekgonul et al. (2023) CBMs remove concept labels altogether. A second thread makes the bottleneck more faithful through probabilistic, energy-based, and Bayesian formulations that model concept uncertainty and dependence Kim et al. (2023); Xu et al. (2024); Feng et al. (2024), and repeatedly confronts *concept leakage*, where unintended information bypasses the named concepts Mahinpei et al. (2021); Sun et al. (2024); this directly motivates our information-bottleneck concept selection, in line with recent IB-based views of CBMs Almudévar et al. (2025). Almost all of this work targets images or text, and extending CBMs to relational graph data remains underexplored. Two concurrent efforts share our name or goal but differ in scope: Xu et al. (2025) augments an *image* CBM with a learned latent concept graph, and Niu et al. (2026) inserts a concept-bottleneck *layer* that reads out global graph-level concepts for graph classification. In contrast, **GCB** targets node classification on text-attributed graphs by aligning a pretrained graph encoder to an LLM-proposed concept space and selecting concepts through an information bottleneck.

**Large language models for text-attributed graphs.** A fast-growing literature couples LLMs with graph learning Jin et al. (2024). LLMs have been used as feature enhancers—TAPE He et al. (2024) distills LLM-generated explanations into node features—as annotators that supply pseudo-labels Chen et al. (2024c), and as zero-shot or instruction-tuned predictors Wang et al. (2024); Chen et al. (2024a); Tang et al. (2024), although when such enhancements actually benefit node classification is itself nuanced Wu et al. (2025). A newer strand turns LLMs toward *interpretability*, producing free-text explanations Pan et al. (2025); Baghershahi et al. (2025) or natural-language rules Armgaan et al. (2025) for a trained GNN. These generate *post-hoc* narratives about a fixed model; **GCB** instead makes LLM-derived concepts the model's intrinsic, intervenable bottleneck, unifying prediction and explanation in a single concept space.

## 3 Graph Concept Bottleneck

Graph Concept Bottleneck consists of three stages: (1) we pretrain a graph-concept mapping model (**Section 3.1**); (2) we create a concept set to explain the target dataset (**Section 3.2** and **Section 3.3**); and (3) we train a predictor that maps concept activations to labels (**Section 3.4**). This multi-stage framework ensures that the model not only achieves strong predictive performance but also produces interpretable, concept-based explanations for its decisions.

### 3.1 Contrastive Concept–Graph Pretraining

We propose Contrastive Concept–Graph Pretraining (CCGP), which pretrains a multimodal model to align graph and text representations in a shared space. CCGP is specifically designed to enhance graph-to-concept alignment and can be applied across diverse datasets and domains.

**Pretraining data.** We collect unlabeled graph data from different domains to construct the pretraining datasets for CCGP. Prior work Wang et al. (2024); Chen et al. (2024a); Tang et al. (2024) has demonstrated the remarkable ability of LLMs to understand and reason over graph-structured data. Motivated by this, we leverage LLMs to generate self-supervised concept annotations. For each dataset, we sample $m$ instances, where each instance $x_i$ corresponds to the ego-network centered at node $v_i$. For each instance $x_i$, we query

GPT-3.5 Brown et al. (2020) to generate a list of associated concepts (see Appendix B.1 for prompt details). We collect instance–concept list pairs $\{(x_i, \mathcal{C}_i)\}$ for future procedures.

We augment the pretrained data to improve the robustness of the pretrained model against data noise. For each instance $x_i$ we create a set of perturbed subgraphs $\mathcal{X}_i^{\text{aug}} = \left\{ \tilde{x}_i^{(1)}, \tilde{x}_i^{(2)}, \ldots, \tilde{x}_i^{(M)} \right\}$, where each $\tilde{x}_i^{(m)}$ is constructed by perturbing the $k$-hop neighborhood of $v_i$ by randomly dropping/adding 20% of the edges. The augmented instance–concept list pairs are obtained as $\{(\mathcal{X}_i^{\text{aug}}, \mathcal{C}_i)\}$

**Encoders.** The pretrained model consists of a graph encoder and a text encoder. The graph encoder $f_\theta^{\text{GNN}}(\cdot)$ with trainable parameter $\theta$ is responsible for capturing both the feature attributes and topological structure of the graph, and it should generalize well to downstream graph data, potentially from different datasets. More expressive architectures, such as Graph Transformers, are capable of modeling rich semantics and complex patterns, but they are more prone to overfitting than smaller GNNs like GCN. We adopt a pretrained Sentence-BERT Reimers & Gurevych (2019) model as the text encoder $f^{\text{LM}}(\cdot)$, with parameters kept frozen throughout training. This choice leverages the model's strong general semantic capabilities, while avoiding the computational cost and overfitting risks associated with fine-tuning large language models on limited data.

**Set2set contrastive learning.** Unlike standard contrastive learning, which aligns the input with a single target, our setting involves multiple augmented graph views and concepts per instance. We thus formulate graph–concept pretraining as a *set-to-set* alignment problem, encouraging the model to consistently associate diverse structural perturbations of a graph with all of its semantic concepts.

For each instance $x_i$, we have a set of augmented views $\mathcal{X}_i^{\text{aug}} = \{\tilde{x}_i^{(1)}, \ldots, \tilde{x}_i^{(M)}\}$ and a concept set $\mathcal{C}_i = \{c_{i1}, c_{i2}, \ldots, c_{iK}\}$. During training, we sample positive pairs from their Cartesian product,

$$P_i = \{(\tilde{x}_i^{(m)}, c_{ij}) \mid m \in \mathcal{M}_i, \ j \in \mathcal{K}_i\},$$

where $\mathcal{M}_i \subseteq \{1, \ldots, M\}$ and $\mathcal{K}_i \subseteq \{1, \ldots, K\}$ are sampled subsets. For each $(\tilde{x}_i^{(m)}, c_{ij}) \in P_i$, we compute the graph embedding $z_i^{(m)} = f_\theta^{\text{GNN}}(\tilde{x}_i^{(m)})$ and the concept embedding $z_{ij}^{\text{concept}} = f^{\text{LM}}(c_{ij})$. We then apply a contrastive loss based on the InfoNCE van den Oord et al. (2018) formulation to maximize the similarity between positive pairs while minimizing similarity to negative pairs in the batch. The contrastive loss for each positive pair is defined as:

$$\mathcal{L}_{i,j}^{(m)} = -\log \frac{\exp\left(\text{sim}\left(z_i^{(m)}, z_{ij}^{\text{concept}}\right)/\tau\right)}{\sum\limits_{(k,l)\in\mathcal{B}} \exp\left(\text{sim}\left(z_i^{(m)}, z_{kl}^{\text{concept}}\right)/\tau\right)}, \tag{1}$$

where $\text{sim}(\cdot, \cdot)$ is cosine similarity, $\tau$ is the temperature, and $\mathcal{B}$ is the set of all (view, concept) pairs in the current batch. Overall, we formulate the learning objective as minimizing the following contrastive loss with respect to $\theta$:

$$\arg\min_\theta \mathcal{L}(\theta) = \frac{1}{\sum_i |\mathcal{M}_i| |\mathcal{K}_i|} \sum_i \sum_{m\in\mathcal{M}_i} \sum_{j\in\mathcal{K}_i} \mathcal{L}_{i,j}^{(m)}(\theta). \tag{2}$$

By doing so, the model learns to align multiple augmented views of each graph with multiple concepts, improving generalization across diverse graph data. We denote the optimized graph encoder as $f^{\text{GNN}}(\cdot)$, whose parameters are frozen during subsequent training to ensure independence from label supervision, thereby preserving semantic meaning and avoiding information leakage.

## 3.2 LLM-Empowered Concept Retrieval

Given the strong ability of LLMs in domain knowledge Lee et al. (2024), abstraction & pattern recognition Lee et al. (2025b), and contextual reasoning Zhang et al. (2024), we construct the concept space through two approaches:

**Global Concept Proposal.** We expect LLMs to identify concepts to distinguish between classes when instructed appropriately. Specifically, we provide a detailed description of the dataset and ask the LLM to generate an initial set of relevant concepts for each class. See B.1 for prompt details.

**Instance-Based Concept Extraction.** While Global Concept Proposal offers broader semantic coverage and reflects domain-level priors, it may overlook dataset-specific nuances that are only accessible through instances. Observing the richness of the training data and LLMs' ability to perform contextual reasoning and summarize fine-grained patterns on graph data Wang et al. (2024); Chen et al. (2024a); Tang et al. (2024), we propose to ask LLMs to recognize relevant concepts given sampled graph instances. Specifically, we sample $m$ graph instances from each class and apply the prompt to each sampled graph instance, resulting in a large set of candidate concepts. We then identify a subset of concepts that are highly relevant to each class, distinct from those used by other classes, and useful for improving class discrimination. To control the quality and size of the concept set we perform several filtering steps to remove redundant or irrelevant concepts. Details of the prompting and filtering process are provided in B.2. We denote the filtered concepts from the Global Concept Proposal and Instance-Based Concept Extraction as $\mathcal{C}^{\text{glob}}$ and $\mathcal{C}^{\text{inst}}$, respectively. We combine $\mathcal{C}^{\text{glob}}$ and $\mathcal{C}^{\text{inst}}$ as the candidate concept set as $\mathcal{C}^{\text{candidate}}$.

### 3.3 Information-Constrained Concept Optimization

The retrieved concept set in Section 3.2 is potentially large, and some of them could be irrelevant or spurious, hindering the explainability and the generalizability of the model. Thus, we adopt the Information Bottleneck (IB) Alemi et al. (2017) criteria to encourage the model to rely on a sparse set of concepts that are most relevant to the prediction.

**Definition 3.1.** The Information Bottleneck criteria is generally formulated as $I(Z; Y) - \beta I(Z; X)$, which seeks a representation $Z$ that is both *informative* and *compressed*: maximizing mutual information with the label $Y$ while minimizing mutual information with the input $X$. A larger $\beta$ results in stronger compression, encouraging $Z$ to retain only the most essential information for predicting $Y$.

In our model, we adopt the IB objective to learn a gating vector $g$ over the fixed concept space. Specifically, for each concept $j$, we learn a soft gate $\mathbf{g}_j$ by applying a sigmoid function to a learnable parameter.

We then apply the gate vector to the concept activation vector of each instance $i$ as $\mathbf{z}_i = \mathbf{g} \odot \mathbf{a}_i$, where $\odot$ is element-wise multiplication, $\mathbf{a}_i$ is the concept activation vector for instance $i$, defined as:

$$\mathbf{a}_i^{(j)} = \text{sim}\left(f^{\text{GNN}}(x_i), \ f^{\text{LM}}(c_j)\right), \tag{3}$$

where $\text{sim}(\cdot, \cdot)$ is the cosine similarity. $\mathbf{z}_i$ is the masked concept vector passed to the classifier. Following the IB principle, we minimize the following objective:

$$\frac{1}{N} \sum_{i=1}^{N} \mathbb{E}\left[-\log q(y_i | \mathbf{z}_i)\right] + \beta \, \text{KL}\left(p(\mathbf{z}_i | \mathbf{a}_i) \, \| \, r(\mathbf{z}_i)\right), \tag{4}$$

where the first term promotes predictive accuracy and the second term minimizes the Kullback–Leibler (KL) divergence between $\mathbf{z}_i$ and $\mathbf{x}_i$, effectively penalizing their mutual information and encouraging a more compressed representation. In our framework, the prediction function $q(y_i | \mathbf{z}_i)$ is parameterized by a trainable multi-layer perceptron $\text{MLP}_\psi^{\text{cls}}$, which takes the masked concept vector $z_i$ as input. The gate vector $\mathbf{g}$, which determines the masking over the concept activations, is computed by a separate network parameterized by $\phi$. Since $\mathbf{z}_i$ is deterministically computed and we do not model a distribution over $p(\mathbf{z}_i | x_i)$, we approximate the KL divergence term with a deterministic sparsity regularizer. In particular, we use an $L_1$ penalty, which encourages the gate values to shrink toward zero, effectively suppressing irrelevant concepts. This results in a sparse, interpretable concept selection, aligning with the Information Bottleneck's objective of compressing the intermediate representation while retaining task-relevant information. Thus, the training objective in Equation 4 becomes:

$$\min_{\phi, \psi} \frac{1}{N} \sum_{i=1}^{N} \mathcal{L}_{\text{CE}}\left(\text{MLP}_\psi^{\text{cls}}(\mathbf{z}_i), \ y_i\right) + \beta \|\mathbf{g}\|_1, \tag{5}$$

where $\mathcal{L}_{\text{CE}}$ denotes the cross-entropy loss between the predicted label distribution and the ground-truth label. While the gates are continuous and soft during training, for interpretability, we require a discrete selection of concepts. To achieve this, after the IB training phase, we freeze the learned gate vector $\mathbf{g}$ and select the top-$K$ concepts with the highest gate values $\mathcal{C}^{\text{selected}} = \text{Top-K}_j(\mathbf{g}_j)$, where $\mathcal{C}^{\text{selected}}$ denotes the final set of selected concepts.

### 3.4 Predictor Learning

We use the selected concept set $\mathcal{C}^{\text{selected}}$ to make the final predictions. For each instance $x_i$, we compute a concept activation vector $\mathbf{a}_i^{\mathcal{C}} \in \mathbb{R}^{|\mathcal{C}^{\text{selected}}|}$ following Equation 3, where each dimension corresponds to a concept in $\mathcal{C}^{\text{selected}}$. We then train a predictor using only the concept activation vector $\mathbf{a}_i^{\mathcal{C}}$ as input. Specifically, we use a classifier $\text{MLP}_\theta^{\text{cls}}$, parameterized by $\theta$ to predict the label $y_i$. The objective is to minimize the loss over the training set:

$$\theta^* = \arg\min_\theta \ \frac{1}{N} \sum_{i=1}^{N} \mathcal{L}_{\text{CE}} \left( \text{MLP}_\theta^{\text{cls}} \left( \mathbf{a}_i^{\mathcal{C}} \right), \ y_i \right), \tag{6}$$

where $\mathcal{L}_{\text{CE}}$ denotes the standard cross-entropy loss, and $N$ is the number of training instances.

## 4 Experiments

Given the self-explainable nature of **GCB**, we evaluate it from two aspects: as a *predictor* and as an *explainer*. Specifically, we investigate whether **GCB** can make robust predictions on text-attributed node classification under distribution shift and data perturbations, and whether the explanations induced by its concepts are faithful to the model's decision process, sufficient and necessary for prediction, and concise enough to be easily interpretable by humans.

Table 1: Node classification performance in *OOD settings* with upsampling ratio $\gamma = 5$. The best-performing interpretable GNN is highlighted in grey, and the overall best-performing method is **bolded**.

| Method | Cora F1 (%) | Cora BACC (%) | Citeseer F1 (%) | Citeseer BACC (%) | Instagram F1 (%) | Instagram BACC (%) | Reddit F1 (%) | Reddit BACC (%) | WikiCS F1 (%) | WikiCS BACC (%) |
|---|---|---|---|---|---|---|---|---|---|---|
| MLP | $50.00_{(0.63)}$ | $60.81_{(0.50)}$ | $38.44_{(0.99)}$ | $54.34_{(1.10)}$ | $35.42_{(0.58)}$ | $51.69_{(0.26)}$ | $16.38_{(0.52)}$ | $51.33_{(0.51)}$ | $54.31_{(0.31)}$ | $65.30_{(0.35)}$ |
| GCN | $55.10_{(0.66)}$ | $64.03_{(1.05)}$ | $46.52_{(0.92)}$ | $59.64_{(0.92)}$ | $36.98_{(0.84)}$ | $50.01_{(0.70)}$ | $12.91_{(0.38)}$ | $48.48_{(0.21)}$ | $59.04_{(1.66)}$ | $\mathbf{68.38}_{(1.73)}$ |
| GAT | $51.30_{(1.27)}$ | $61.52_{(1.15)}$ | $45.62_{(1.01)}$ | $58.77_{(0.87)}$ | $33.31_{(0.59)}$ | $50.18_{(0.41)}$ | $12.93_{(0.30)}$ | $49.34_{(0.18)}$ | $57.05_{(1.00)}$ | $64.53_{(0.85)}$ |
| SAGE | $44.26_{(1.78)}$ | $53.95_{(1.65)}$ | $30.87_{(0.41)}$ | $48.42_{(0.47)}$ | $31.46_{(0.20)}$ | $48.27_{(0.18)}$ | $13.38_{(0.17)}$ | $49.18_{(0.47)}$ | $51.87_{(1.24)}$ | $62.06_{(1.37)}$ |
| GT | $38.26_{(1.64)}$ | $48.66_{(1.36)}$ | $28.38_{(1.58)}$ | $48.22_{(0.77)}$ | $30.90_{(0.43)}$ | $48.06_{(0.37)}$ | $12.64_{(0.54)}$ | $48.62_{(0.26)}$ | $54.06_{(0.84)}$ | $62.50_{(0.77)}$ |
| DIR-GNN | $23.07_{(2.70)}$ | $43.18_{(2.13)}$ | $15.31_{(1.33)}$ | $42.93_{(1.00)}$ | $26.74_{(0.00)}$ | $50.00_{(0.00)}$ | $8.46_{(0.00)}$ | $50.00_{(0.00)}$ | $22.93_{(1.35)}$ | $42.11_{(0.42)}$ |
| GIB | $19.23_{(4.10)}$ | $40.24_{(3.48)}$ | $15.52_{(1.79)}$ | $42.31_{(1.19)}$ | $26.75_{(0.01)}$ | $50.01_{(0.01)}$ | $8.47_{(0.03)}$ | $50.01_{(0.01)}$ | $24.98_{(1.27)}$ | $39.15_{(1.12)}$ |
| VGIB | $44.56_{(6.43)}$ | $57.06_{(4.66)}$ | $22.26_{(6.35)}$ | $47.72_{(3.26)}$ | $26.74_{(0.00)}$ | $50.00_{(0.00)}$ | $8.46_{(0.00)}$ | $50.00_{(0.00)}$ | $56.02_{(1.76)}$ | $64.14_{(1.25)}$ |
| SEGNN | $30.68_{(2.91)}$ | $48.75_{(1.96)}$ | $19.92_{(2.80)}$ | $42.89_{(1.55)}$ | $26.74_{(0.00)}$ | $50.00_{(0.00)}$ | $8.46_{(0.00)}$ | $50.00_{(0.00)}$ | $34.97_{(1.26)}$ | $50.71_{(1.01)}$ |
| **GCB** | $\mathbf{56.63}_{(1.38)}$ | $\mathbf{66.71}_{(0.99)}$ | $\mathbf{60.19}_{(0.61)}$ | $\mathbf{67.12}_{(0.60)}$ | $\mathbf{56.80}_{(0.23)}$ | $\mathbf{58.47}_{(0.38)}$ | $\mathbf{48.16}_{(0.25)}$ | $\mathbf{63.07}_{(0.99)}$ | $56.36_{(0.46)}$ | $67.57_{(0.73)}$ |

Table 2: Node classification performance in *perturbation settings* with pertubation ratio $\rho = 0.3$. The best-performing interpretable GNN is highlighted in grey, and the overall best-performing method is **bolded**.

| Method | Cora F1 (%) | Cora BACC (%) | Citeseer F1 (%) | Citeseer BACC (%) | Instagram F1 (%) | Instagram BACC (%) | Reddit F1 (%) | Reddit BACC (%) | WikiCS F1 (%) | WikiCS BACC (%) |
|---|---|---|---|---|---|---|---|---|---|---|
| MLP | $43.08_{(0.45)}$ | $56.55_{(0.37)}$ | $37.77_{(0.53)}$ | $53.77_{(0.59)}$ | $36.69_{(0.45)}$ | $52.57_{(0.19)}$ | $16.75_{(0.84)}$ | $51.11_{(0.44)}$ | $53.70_{(0.32)}$ | $64.95_{(0.45)}$ |
| GCN | $56.21_{(1.25)}$ | $66.49_{(0.85)}$ | $46.45_{(1.42)}$ | $58.14_{(1.61)}$ | $43.87_{(3.28)}$ | $54.11_{(0.85)}$ | $16.64_{(0.77)}$ | $50.43_{(0.76)}$ | $61.45_{(0.38)}$ | $65.64_{(0.87)}$ |
| GAT | $52.64_{(1.33)}$ | $61.70_{(0.71)}$ | $46.55_{(0.87)}$ | $60.54_{(0.58)}$ | $37.07_{(1.09)}$ | $52.38_{(0.43)}$ | $19.23_{(1.76)}$ | $52.20_{(0.84)}$ | $59.34_{(0.90)}$ | $66.75_{(1.21)}$ |
| SAGE | $53.15_{(2.06)}$ | $57.70_{(2.21)}$ | $39.37_{(1.30)}$ | $56.60_{(0.90)}$ | $38.45_{(2.14)}$ | $52.79_{(0.90)}$ | $16.51_{(0.49)}$ | $51.64_{(0.39)}$ | $61.58_{(0.44)}$ | $60.76_{(0.49)}$ |
| GT | $46.13_{(2.23)}$ | $55.15_{(1.80)}$ | $33.36_{(1.69)}$ | $52.65_{(0.93)}$ | $35.36_{(0.78)}$ | $51.70_{(0.27)}$ | $17.29_{(0.50)}$ | $51.31_{(0.12)}$ | $57.88_{(0.95)}$ | $62.61_{(1.60)}$ |
| DIR-GNN | $70.78_{(2.43)}$ | $70.20_{(2.49)}$ | $62.03_{(0.86)}$ | $\mathbf{64.65}_{(0.70)}$ | $55.56_{(1.43)}$ | $56.45_{(0.64)}$ | $54.13_{(1.52)}$ | $\mathbf{56.35}_{(0.57)}$ | $57.07_{(3.45)}$ | $56.34_{(1.97)}$ |
| GIB | $32.94_{(18.33)}$ | $37.41_{(15.43)}$ | $47.23_{(15.64)}$ | $52.91_{(11.58)}$ | $38.55_{(6.36)}$ | $50.74_{(0.95)}$ | $39.96_{(7.82)}$ | $51.54_{(1.85)}$ | $21.62_{(10.37)}$ | $25.78_{(9.43)}$ |
| VGIB | $20.15_{(26.58)}$ | $26.24_{(23.92)}$ | $54.92_{(20.60)}$ | $57.43_{(17.87)}$ | $39.13_{(0.61)}$ | $50.09_{(0.17)}$ | $34.79_{(3.10)}$ | $50.20_{(0.39)}$ | $58.67_{(24.39)}$ | $59.70_{(22.48)}$ |
| SEGNN | $52.58_{(4.71)}$ | $56.78_{(3.34)}$ | $59.76_{(1.11)}$ | $62.69_{(1.06)}$ | $55.15_{(0.64)}$ | $55.40_{(0.43)}$ | $55.44_{(0.85)}$ | $55.77_{(0.71)}$ | $38.08_{(1.10)}$ | $42.11_{(1.15)}$ |
| **GCB** | $\mathbf{70.98}_{(0.73)}$ | $\mathbf{71.36}_{(1.07)}$ | $\mathbf{63.44}_{(0.29)}$ | $63.84_{(0.32)}$ | $\mathbf{56.65}_{(0.26)}$ | $\mathbf{56.61}_{(0.27)}$ | $\mathbf{55.56}_{(0.74)}$ | $55.58_{(0.77)}$ | $\mathbf{66.08}_{(0.53)}$ | $\mathbf{70.43}_{(0.75)}$ |

### 4.1 Datasets

Following the practice in GraphCLIP (Zhu et al., 2025), we use *non-overlapping* text-attributed graphs from diverse domains to pretrain the Graph-Concept Alignment model. The graph data used for pre-training is required to be of the same type as the downstream datasets to ensure transferability.

**Source datasets.** We use five source datasets for CCGP: `Pubmed` (Sen et al., 2008) is a citation network in the medical domain. `Ele-Computers`, `Sports-Fitness`, `Books-Children`, and `Books-History` (Yan et al., 2023b) are e-commerce co-purchasing networks. For each dataset, we sample 1,000 nodes and query GPT-3.5

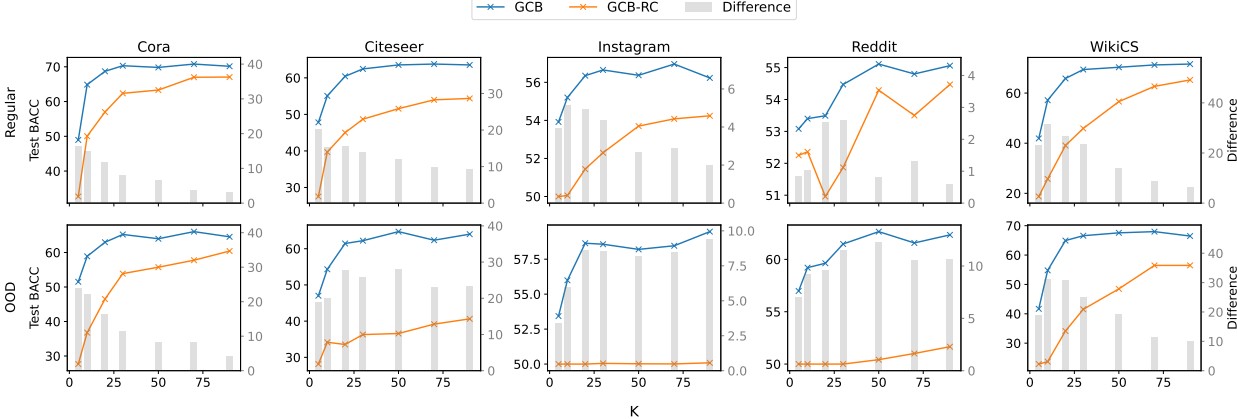

Figure 1: Performance of the original **GCB** compared to its variant with random concepts (GCB-RC) across different concept set sizes, on regular splits (top row) and OOD splits (bottom row).

to generate 10 concepts that appear in each node's ego network, serving as labels for the graph-concept alignment.

**Target datasets.** We use `Cora` (Sen et al., 2008), `Citeseer` (Sen et al., 2008), `Instagram` (Huang et al., 2024), `Reddit` (Huang et al., 2024), and `WikiCS` (Mernyei & Cangea, 2020) as target datasets. `Cora` and `Citeseer` are citation networks in Computer Science; `Instagram` and `Reddit` are social networks; and `WikiCS` is a Wikipedia article network. *We ensure that all target datasets are from different domains than the source datasets to evaluate model generalizability and reduce data leakage.* We evaluate them under three settings:

- *Regular.* We randomly split data into training, validation, and test sets that follow the same data distribution.
- *OOD.* Following the *soft label-leaveout* strategy (Han et al., 2025), we create mismatched class distributions across splits using an upsampling ratio $\gamma$. Specifically, classes are partitioned into majority and minority groups. Instances from majority classes are then $\gamma$ times more likely to be sampled into the training/validation set than those from minority classes, while keeping the overall training/validation/test split ratios unchanged. A larger $\gamma$ therefore induces a stronger distribution shift between the training and test sets. We set $\gamma \in \{2, 3, 5, 10\}$; the special case $\gamma=1$ (no upsampling) recovers the IID split and serves as the in-distribution anchor in our shift-intensity analysis (Figure 5).
- *Adversarial.* Using the same split as the *regular setting*, we perturb the edges in the training set by randomly dropping and adding edges to the graph with a perturbation ratio $\rho$. We set $\rho \in \{0.05, 0.1, 0.2, 0.3, 0.5\}$.

We report the Macro-F1 and Balanced Accuracy (BACC) scores to evaluate model performance to account for class imbalance. See C.1 and C.2 for further details on the datasets and experimental settings.

## 4.2 GCB as a Predictor

We evaluate **GCB** as a predictive model on target datasets under three different settings. We set the default number of concepts of **GCB** $K$ as 30. For each setting, we also evaluate a set of SOTA GNN and MLP models, including MLP, GCN Kipf & Welling (2017), GAT Veličković et al. (2018), GraphSAGE (SAGE) Hamilton et al. (2017), and Graph Transformer (GT) Yun et al. (2019), as baselines. We also compare with self-explainable GNNs including GIB Yu et al. (2021), VGIB Yu et al. (2022), DIR-GNN Wu et al. (2022), and SEGNN Dai & Wang (2021).

*(1)* ***GCB*** *improves the model generalizability in OOD data.* We evaluate **GCB** under the *OOD setting* across different upsampling ratios. We report the test results in Table 1 for the upsampling ratio $\gamma = 5$. We conduct ablation study on the effect of distribution shift intensity in Section 4.4.2. The results show that **GCB** not only significantly outperforms all self-explainable graph learning methods, but also consistently surpasses or is comparable to SOTA GNNs. We attribute this to **GCB**'s reliance on a constrained set of

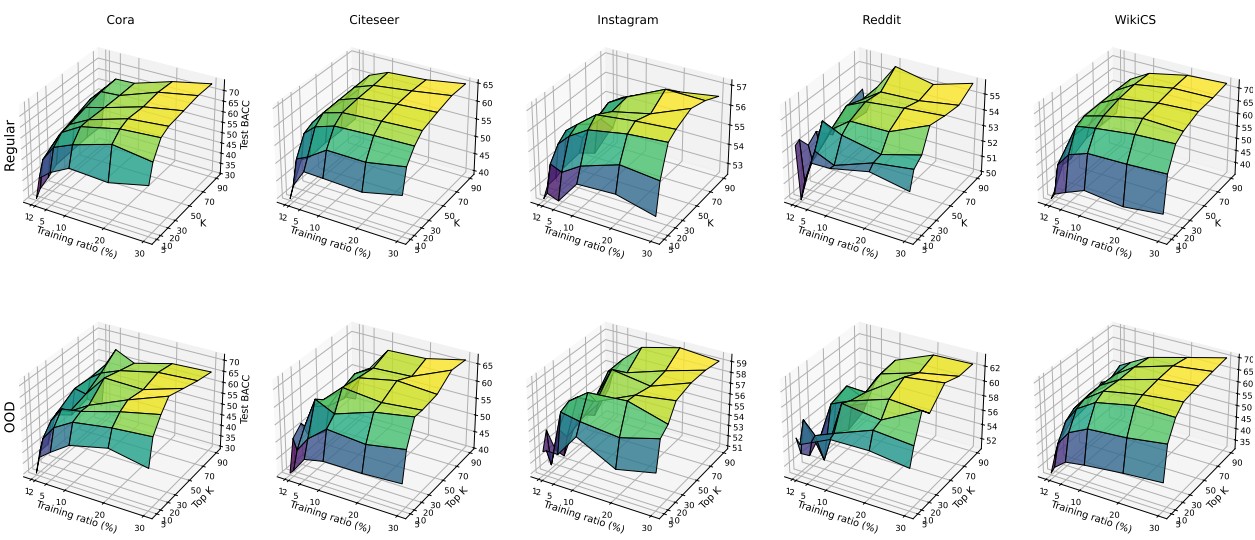

Figure 2: Performance of **GCB** across different concept sizes ($K$) and training ratios (%) on regular splits and OOD splits.

concepts for prediction, which potentially contains fewer spurious features and thus makes it less susceptible to distribution shifts. **GCB** is therefore a strong baseline for improving OOD generalizability in graph learning.

*(2) **GCB** improves model robustness under data perturbations.* We evaluate **GCB** under the *adversarial setting* with different perturbation ratios. We only report the test results in Table 2 for perturbation ratio $\rho = 0.3$. We conduct ablation study on the effect of pertubation intensity in Section 4.4.3. We observe that while most GNNs perform well under clean conditions, their performance degrades significantly when trained on perturbed data, highlighting their vulnerability to poisoning attacks. In contrast, **GCB** demonstrates strong robustness against perturbed train data, while maintaining performance comparable to the model trained on clean data. We attribute this robustness to the use of a pretrained graph encoder trained on augmented data from diverse domains.

*(3) **GCB** incurs minimal cost in model utility on clean in-distribution data.* We evaluate **GCB** and baseline methods under the *regular setting*, and report the test BACC scores (averaged over 5 trials) in Table 6. On three out of five datasets, **GCB** achieves the best performance (in at least one metric) among interpretable GNN methods. Moreover, compared to the overall best-performing model, **GCB** delivers competitive results with only small performance gaps, demonstrating its ability to retain high predictive utility while offering interpretability.

### 4.3 GCB as an Explainer

We now examine the quality of the interpretation offered by **GCB**. Specifically, we evaluate it in the following terms:

- *Faithfulness.* Is it meaningful and relevant?
- *Necessity* and *sufficiency.* How many concepts actually carry each decision?
- *Sparsity.* Is it concise and human-tractable?

Together, they assess whether the concepts are functionally responsible for the model's decisions and whether the resulting explanations are concise, faithful, and meaningful.

(1) **Faithfulness.** While the quality of the explanation and prediction performance are correlated due to its self-explainable nature, there is a concern about information leakage, which can undermine the faithfulness Havasi et al. (2022); Sun et al. (2024). When the label predictor is trained, it might exploit spurious signals from the concept activations produced by the concept predictor rather than relying on the

semantics of the concepts. In such cases, even if the concepts are meaningless or unrelated, the model could still achieve high accuracy by assigning higher activation scores to arbitrary concepts that correlate with the label, providing no true explainable value. Thus, we want to answer a crucial question: *Is the accurate prediction provided by **GCB** driven by a faithful projection of the input into a semantic space that is genuinely relevant to the label, or is it merely the result of information leakage?* Inspired by Mahinpei et al. (2021), since there is no easy way to directly measure information leakage, we assess it indirectly by replacing the retrieved concepts with random ones. Intuitively, *if the model maintains strong performance with meaningless concepts, this indicates that its predictions do not truly rely on the intended concept subspace, but instead exploit alternative pathways, suggesting the presence of information leakage.* To eliminate any real-world semantics, we construct a set of random concepts consisting of arbitrary tokens representing random numbers (e.g., "1", "99", etc.). We report the performance of **GCB** using both retrieved concepts ("GCB") and random concepts ("GCB-RC") across different numbers of selected concepts $K$, under *regular* and *OOD* settings in Figure 1. The difference between the two plotted as a gray bar can measure the degree of information leakage. For the regular split, we observe a general pattern: the performance gap gradually decreases as the concept size increases. Specifically, GCB-RC performs significantly worse with smaller concept sizes but gradually approaches GCB's performance as the concept size grows. This suggests that when the concept set is large enough, the model may rely more on spurious correlations between concept activation patterns and labels. Conversely, when the concept set is small, the spurious patterns are harder to explore, and the relevance of actual concepts plays a more critical role. For the OOD split, random concepts fail across all concept sizes, highlighting the inability of random concepts to generalize beyond the training distribution. These findings indicate that although random concepts can achieve reasonable performance with a sufficiently large concept set under in-distribution data, they fail when the concept set is limited or when distribution shifts occur. This highlights the importance of carefully controlling the size of the concept set: *using a small, curated set of concepts not only enhances the interpretability of the model, but also helps ensure the quality and faithfulness of its explanations.*

(2) **Necessity and sufficiency.** A good explanation should be both *necessary* and *sufficient.* Using too many concepts can hurt interpretability and, as discussed in the previous section, make the model more susceptible to information leakage, while using too few may deprive the model of essential information and impair predictive accuracy. We therefore examine the sensitivity of **GCB** to the concept size $K$ under varying training ratios for each dataset. The results are visualized in Figure 2, where the x-axis denotes the training ratio and the y-axis indicates the number of concepts. Across datasets, we observe a consistent pattern: *performance improves rapidly as $K$ increases, but the gains gradually diminish and eventually plateau.* This indicates that a relatively small number of concepts is sufficient to capture most of the predictive signal. In the OOD setting, however, enlarging the concept set can even degrade performance, particularly when the training ratio is small. Having too many concepts appears to hinder generalization, likely by introducing spurious or unstable signals. Overall, this suggests that a moderate concept size offers the best trade-off between expressiveness and robustness. In practice, selecting $K$ in the range of 30–50 achieves strong performance while preserving the necessity and sufficiency of the explanations.

(3) **Sparsity.** We study whether **GCB** can provide concise and human-tractable explanations via a case study on `Cora` with two sampled instances as shown in Figure 3. For each instance, we visualize the concept activations as a heatmap: for compactness we display the top-10 concepts, formed by taking the union of the most-activated concepts from each encoder. In each cell, the number indicates the raw activation score, while the color encodes the concept's contribution—its share of the instance's total activation—so the colorbar is scaled by ratio rather than absolute magnitude (darker indicates a larger share). We compare activations produced by an MLP encoder (feature-only) and a GCN encoder (with 2-hop neighborhoods) to reveal the role of graph structure. In the upper instance, the MLP highlights *Genetic Programming, Simulated Evolution, and Genetic Operators*, but also activates less relevant concepts such as *Synthetic Maze Tasks* and *Bayesian Networks.* In contrast, the GCN concentrates on the most pertinent evolutionary concepts, indicating that neighborhood information helps sharpen semantic focus. Similarly, for the lower instance, the MLP emphasizes *Bayesian Networks* and *Probabilistic Graphical Models* yet also activates *Genetic Programming*, whereas the GCN primarily activates highly relevant concepts such as *Hidden Markov Models* and *Bayesian Inference.* These sharper activations translate into better predictions, shown in the adjacent class-probability heatmap over Cora's seven classes (with the ground-truth class starred). For the upper instance, both

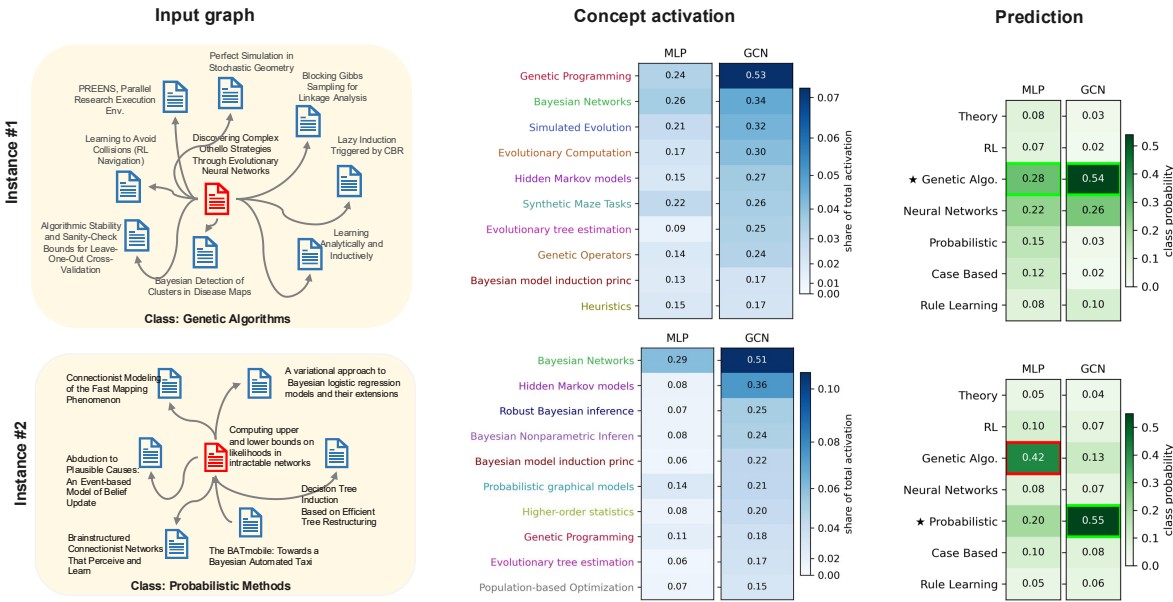

Figure 3: Case study on Cora. We visualize the concept activations of two instances produced by both the MLP encoder, which uses only node features, and the GCN encoder, which incorporates 2-hop neighborhood information. For each instance (row): the input graph shows the target paper (red) and its neighbors (blue); the concept activation panel shows, for a shared set of top concepts (the union of each encoder's most-activated concepts), each concept's share of the instance's total activation (normalized to sum to one, darker = larger share); and the prediction panel shows the class distribution over Cora's seven classes. The GCN concentrates activation on the relevant concepts and predicts more confidently — correctly classifying Instance #2, which the MLP misclassifies.

encoders correctly predict *Genetic Algorithms*, but the GCN does so with markedly higher confidence (0.54 vs. 0.28). For the lower instance, the MLP misclassifies the node as *Genetic Algorithms* (0.42), whereas the GCN correctly predicts *Probabilistic Methods* (0.55). These comparisons demonstrate that incorporating graph structure yields more accurate concept activations and, in turn, more accurate predictions. We also emphasize that the explanation is not determined by only a few top-activated concepts, but by the entire activation landscape across all concepts. When the concept activation is properly visualized—such as through a heatmap—it enables users to gain intuitive insights into the model's decision process that go beyond what can be conveyed by natural language alone.

## 4.4 Ablation study

We conduct ablation studies to further understand the design choices and robustness behavior of **GCB**. First, we examine the effect of different graph encoders in the graph-concept alignment module (Section 4.4.1). Second, we analyze how **GCB**'s advantage over GCN changes as the intensity of distribution shift increases, using Jensen-Shannon divergence to quantify the train-test mismatch (Section 4.4.2). Third, we study the impact of adversarial edge perturbations to evaluate whether **GCB**'s robustness depends on the magnitude of structural noise (Section 4.4.3).

### 4.4.1 Graph encoders.

We study how different graph-text alignment models affect performance. First, we compare versions of **GCB** using different graph encoders: GCN (default), GAT, and Graph Transformer (GT). We also explore the effect

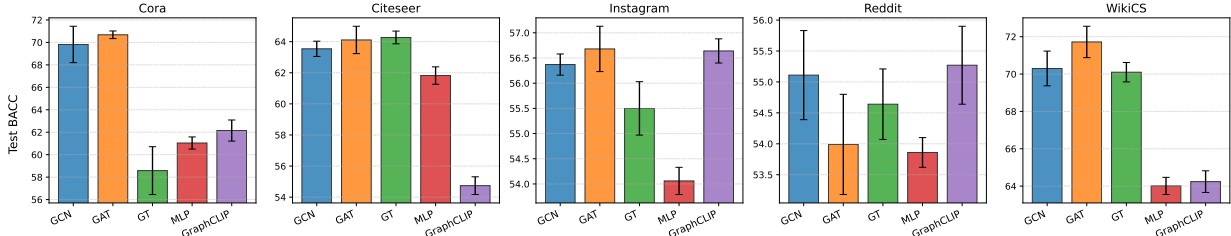

Figure 4: Performance of **GCB** variations using different graph encoders.

of removing the graph structure by replacing the graph encoder with an MLP for decoding the concept map. Additionally, we evaluate a pretrained graph foundation model, GraphCLIP, which includes both graph and text encoders for graph-text alignment. All results are shown in Figure 4. We observe that GCN consistently performs well across all datasets compared to GAT and GT, suggesting that a simpler architecture may be more stable when pretraining data is limited. The model's performance drops significantly when using the MLP encoder, *highlighting the importance of leveraging graph structure for mapping input graphs into the concept space.* GraphCLIP performs slightly better on `Instagram` and `Reddit` but considerably worse on the other three datasets. One potential reason is that GraphCLIP aligns graphs to free-form summaries that contain noisy information, which can lead to inaccurate mappings between graphs and concepts. Moreover, GraphCLIP jointly trains both the graph and text encoders and may cause overfitting, especially when the training corpus is small or domain-specific. This limitation could explain why GraphCLIP performs well on `Instagram` and `Reddit`—social networks that likely share overlapping topics with its training corpus—but poorly on `Cora`, `Citeseer`, and `WikiCS`.

### 4.4.2 Effect of Distribution-Shift Intensity

We investigate how the intensity of distribution shift affects the model's performance. We sweep the OOD upsampling ratio $\gamma$, and ask how **GCB**'s gain over the GCN baseline *scales* with the intensity of the shift, rather than merely with its presence. Throughout this subsection we report the *improvement*

$$\Delta \;=\; \text{score}(\textbf{GCB}) - \text{score}(\text{GCN}) \quad (\%),$$

because the OOD test split itself depends on $\gamma$ (see Section 4.1). Although a larger $\gamma$ increases the train-test mismatch, it may also rebalance the test set. As a result, a larger $\gamma$ does not necessarily lead to lower prediction accuracy. Since both methods are evaluated on the identical split at each $\gamma$, plotting $\Delta$ partially factors out this split-induced confounding effect and better isolates the contribution of the model itself.

**Measuring the intensity of the OOD shift.** Let $P$ be the class distribution over the training pool $(\text{train} \cup \text{valid})$ and $Q$ be the class distribution over the test set. We summarize the train/test mismatch via the Jensen–Shannon divergence (in bits)

$$\text{JSD}(P, Q) \;=\; H\!\left(\tfrac{P+Q}{2}\right) - \tfrac{1}{2}H(P) - \tfrac{1}{2}H(Q),$$

where $H(\cdot)$ is the Shannon entropy. JSD is symmetric, bounded in $[0, 1]$, takes the value 0 for identical distributions and 1 for disjoint supports. We use JSD rather than $|H(P) - H(Q)|$ because entropy measures only *balance*: two distributions can have identical entropy yet disjoint supports. For each (dataset, $\gamma$) pair we reproduce the soft-label-leaveout split for 5 random seeds and report the mean JSD; the IID setting ($\gamma{=}1$) yields JSD close to 0 on every dataset.

**Effect of distribution-shift intensity.** Figure 5 reveals the following patterns. *(i)* At the IID anchor (JSD$\approx$0), $\Delta$ is small in absolute value on every dataset (within $\pm 2$ pp), confirming that **GCB** pays essentially no in-distribution accuracy cost. *(ii)* As JSD grows, $\Delta$ widens markedly on the four shift-sensitive datasets. `Reddit` shows the steepest gradient ($\Delta$F1 climbs from +0.6 pp at IID to +27–+35 pp once JSD $> 0.15$); `Citeseer` and `Instagram` follow similar trajectories, reaching $\Delta$F1 of +13 to +20 pp under stronger shift;

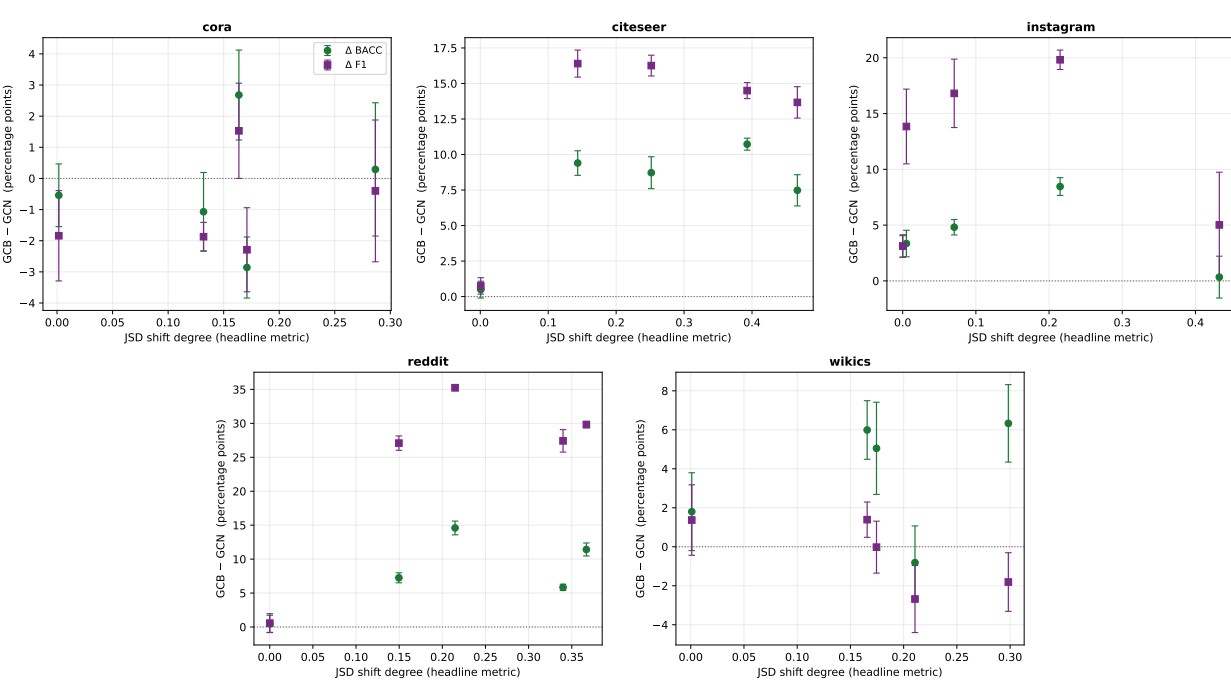

Figure 5: **GCB**'s improvement over GCN ($\Delta = \textbf{GCB} - \text{GCN}$, percentage points) as a function of the train/test distribution-shift intensity measured by JSD. Each panel is one target dataset; the five points per panel correspond to $\gamma \in \{1, 2, 3, 5, 10\}$, with $\gamma=1$ giving the IID anchor at JSD$\approx$0. Both BACC ($\circ$) and Macro-F1 ($\square$) improvements are reported; error bars are over 5 random seeds.

`WikiCS` exhibits a moderate but consistent BACC gain of $\sim +5$–$+6$ pp. *(iii)* `Cora` is the lone exception: GCN is already a strong baseline on this small, label-balanced citation graph, and $\Delta$ stays near zero across all $\gamma$. Overall, the gain of **GCB** is *not uniform*—it tracks the magnitude of the shift—which directly supports the claim that **GCB**'s constrained concept bottleneck reduces reliance on spurious features that fail to transfer under shifted test distributions.

### 4.4.3 Effect of Perturbation Magnitude

We also sweep the adversarial perturbation ratio $\rho$. Figure 6 tells a qualitatively different story. Unlike the OOD trend, where $\Delta$ grows monotonically with the shift intensity, here $\Delta$ is approximately *step-like*: it is near zero at $\rho=0$ and then jumps to a high plateau as soon as the graph is perturbed, staying roughly flat across $\rho \in \{0.05, \dots, 0.5\}$. On `Reddit`, $\Delta\text{F1} \approx +38$ pp uniformly across all $\rho > 0$; on `Citeseer`, $\Delta\text{F1} \approx +16$–$+17$ pp; `Instagram` sits around $+13$–$+19$ pp; `Cora` and `WikiCS` show smaller but consistently positive gains. The step-like shape indicates that GCN's failure is essentially *binary*—any nonzero structural noise breaks its message passing—while **GCB**'s frozen, pretrained graph encoder absorbs the perturbation almost independently of its magnitude. This complements the OOD finding: **GCB**'s robustness arises from two distinct mechanisms (concept-space regularization under distribution shift, and a pretrained encoder under structural noise), and the two robustness profiles look quantitatively different.

### 4.5 Cross-LLM Analysis

**GCB** relies on an LLM to propose the candidate concept space (Section 3.2), which raises a natural question: is the resulting concept bottleneck an artifact of one particular LLM, or do independent LLMs converge on the same concepts? To answer this question, we run the concept-generation pipeline of Section 3.2 on all five datasets with five LLMs spanning distinct families and parameter scales: GPT-3.5, DeepSeek-V3.1 DeepSeek-AI et al. (2024) (671B), Qwen-3.5 (397B), Gemma-3 (27B), and Mistral-Large-3 (675B). Each model receives

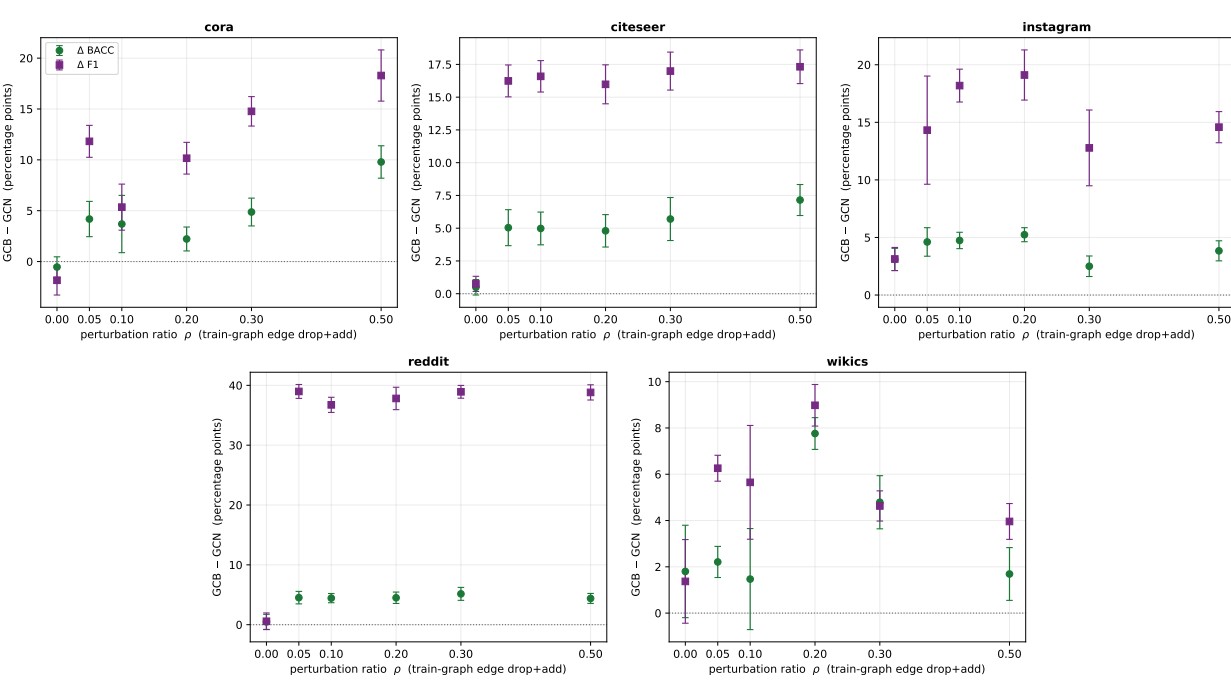

Figure 6: **GCB**'s improvement over GCN as a function of the adversarial perturbation ratio $\rho$. Each panel is one target dataset; $\rho{=}0$ is the clean anchor (identical split, no edges perturbed), and $\rho{\in}\{0.05, 0.1, 0.2, 0.3, 0.5\}$ correspond to random edge drop/add on the training graph. Same axes and markers as Figure 5.

the same prompts and passes through the same filtering pipeline, yielding a finalized concept set on each dataset. We then ask how much two models agree in terms of exact wording and semantic meaning.

**Measuring concept similarity**. For each ordered pair of models $(A, B)$ and each concept $c \in \mathcal{C}_A$, we find its best match in $\mathcal{C}_B$ under three increasingly lenient notions of similarity, plus a fourth measure that operates at the class level. *(i) Exact-string*: the fraction of concepts in $\mathcal{C}_A$ that appear verbatim (case-insensitive) in $\mathcal{C}_B$. *(ii) Lexical*: the maximum token-level Jaccard overlap $\max_{c' \in \mathcal{C}_B} \frac{|t(c) \cap t(c')|}{|t(c) \cup t(c')|}$, where $t(\cdot)$ is the set of word tokens. *(iii) Semantic*: the maximum cosine similarity $\max_{c' \in \mathcal{C}_B} \cos(\phi(c), \phi(c'))$ between SBERT sentence embeddings $\phi(\cdot)$ (all-MiniLM-L6-v2). *(iv) Per-class centroid*: we assign each concept to its nearest class label, average the embeddings of a model's concepts within each class to obtain a per-model class centroid, and take the cosine between the two models' centroids for the same class. The first two capture surface form, the third the meaning of individual concepts, and the fourth the meaning of the per-class concept geometry. Each metric is averaged over all 20 ordered model pairs (the centroid additionally over the classes).

Table 3: Cross-LLM concept agreement on all five datasets, each averaged over the 20 ordered model pairs. Surface metrics (exact, lexical) are low everywhere; semantic agreement is high on the academic/topic graphs but markedly lower on the social networks. The per-class centroid geometry stays shared throughout. Exact is a percentage; the rest are cosine ($\phi$) or token-Jaccard.

| Dataset | Exact (%) | Lexical (Jacc.) | Semantic (cos) | Centroid (cos) |
|---|---|---|---|---|
| Cora | 26.3 | 0.47 | 0.80 | 0.94 |
| Citeseer | 26.6 | 0.46 | 0.79 | 0.89 |
| WikiCS | 27.3 | 0.46 | 0.80 | 0.94 |
| Instagram | 4.9 | 0.21 | 0.67 | 0.92 |
| Reddit | 3.0 | 0.14 | 0.62 | 0.88 |

Table 3 shows that surface agreement is low on every dataset—at most 27% exact and 0.47 lexical overlap—so no two LLMs phrase concepts the same way. What matters for **GCB**, however, is not the wording but the *class-level concept geometry* the bottleneck consumes, and there the models agree closely and uniformly: the per-class centroid cosine is high on all five datasets (0.88–0.94), academic and social graphs alike. Individual-concept semantic agreement tracks how canonical the domain vocabulary is—it is very high on the academic/topic graphs (mean best-match cosine ≈ 0.80 on `Cora`, `Citeseer`, `WikiCS`) and more moderate on the social networks (0.62–0.67 on `Instagram`, `Reddit`), where notions such as posting behavior and user roles admit more phrasings. But this affects only the granularity of individual concepts: every LLM still recovers the same per-class concept space, so the convergence holds regardless of graph type.

To calibrate the cosine scale we compare against three references on each dataset: a random cross-model pair scores 0.34–0.39 (the background floor), a perfect identity match scores 1.00, and a model's *own* nearest neighbor—the closest distinct concept within the same set—scores 0.66–0.73. On all five datasets the cross-model nearest neighbor is at least as tight as a model's internal nearest neighbor and far above the floor: because each LLM de-duplicates its own concept list, its internal neighbor is pushed apart, whereas a different LLM freely re-expresses the same idea in new words. Notably, the *per-class* concept geometry stays shared even where individual concepts diverge—averaging concept embeddings within each class and comparing centroids across models yields a mean cosine of 0.88–0.94 on every dataset, social networks included.

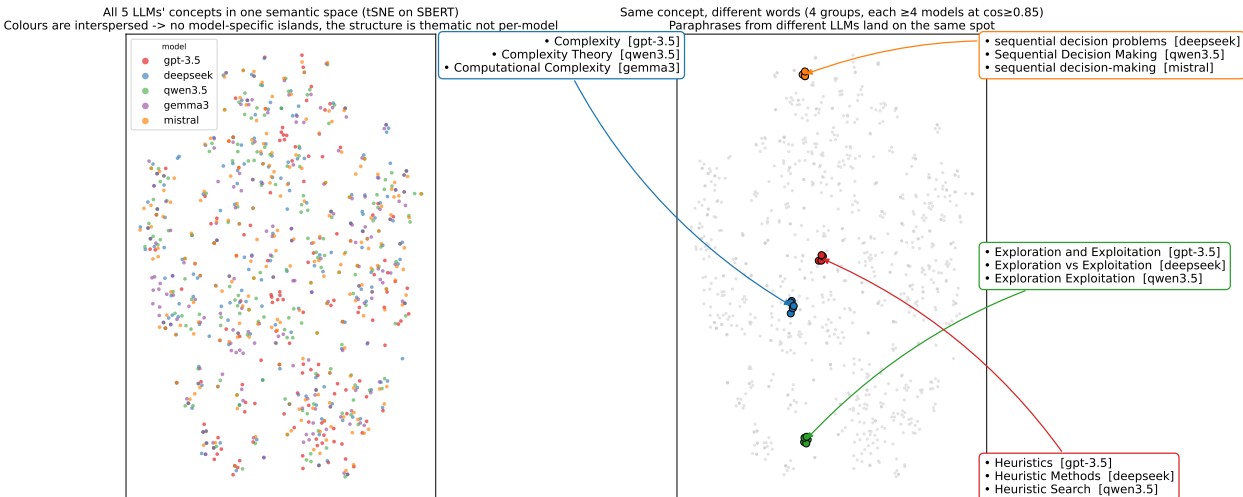

Figure 7: Concept convergence across five LLMs on `Cora` (representative of the academic graphs; `Citeseer` and `WikiCS` look the same, while the social graphs spread individual concepts more widely but keep the same shared per-class structure): all 866 finalized concepts projected to 2D with t-SNE over SBERT embeddings. *Left:* points colored by their source LLM; the colors are fully interspersed with no model-specific islands, so the five concept clouds occupy one shared semantic space. *Right:* representative paraphrase groups in which four or five different LLMs land on the same location with different wordings (e.g., *Complexity/Computational Complexity*, *Exploration and Exploitation/Exploration vs Exploitation*, *Heuristics/Heuristic Search*).

Figure 7 projects all 866 finalized `Cora` concepts (pooled across the five LLMs) into 2D with t-SNE over SBERT embeddings. In the left panel the points are colored by source LLM and are fully interspersed—there are no model-specific islands—indicating one shared semantic space rather than five disjoint ones. The right panel highlights paraphrase groups in which four or five different LLMs land on the same spot with different surface strings, making the convergence concrete. `Citeseer` and `WikiCS` produce the same interspersed picture (with paraphrase groups such as *Machine Learning/Supervised Learning* and *Web framework/Web application framework*); on `Instagram` and `Reddit` the individual concepts spread more widely, yet they remain interspersed by model and aligned at the class level (per-class centroid 0.88–0.92 in Table 3), so the concept space stays shared across LLMs.

**Downstream performance.** The analysis so far measures convergence on the concepts themselves; we now verify that it carries through to **GCB**'s predictions. We re-run the full **GCB** pipeline five times on each

dataset—each run seeded with the concept space proposed by a different LLM—and report the resulting node-classification accuracy (best $\beta$ per run, averaged over five seeds) in Table 4. The choice of LLM barely moves the result: on every dataset the five LLMs land within 1.5 accuracy points of one another (range 0.50–1.45, cross-LLM standard deviation 0.19–0.52 points), with largely overlapping confidence intervals. In contrast to the individual-concept semantic agreement—high on the academic graphs, more moderate on the social ones—this performance agreement is uniform across graph types (mean cross-LLM standard deviation 0.33 on academic vs. 0.35 on social): even where different LLMs phrase social-behavior concepts more diversely, they induce the same class-level geometry (Table 3) and therefore the same downstream accuracy. **GCB**'s predictive quality is thus a property of the task rather than of the LLM that seeds its concept space.

Table 4: Downstream **GCB** node-classification accuracy (%) when the candidate concept space is proposed by each of the five LLMs (GCN encoder, best $\beta$). Each cell is the mean $\pm$ standard deviation over five seeds. The accuracy is nearly identical across LLMs on every dataset; *Range* is the maximum$-$minimum of the five mean accuracies. The agreement is uniform across academic and social graphs, unlike the individual-concept semantic agreement in Table 3.

| Dataset | GPT-3.5 | DeepSeek | Qwen-3.5 | Gemma-3 | Mistral-L | Range |
|---------|---------|----------|----------|---------|-----------|-------|
| Cora | $83.51_{\pm 0.69}$ | $82.95_{\pm 0.92}$ | $82.06_{\pm 1.40}$ | $82.24_{\pm 1.06}$ | $82.83_{\pm 1.28}$ | 1.45 |
| Citeseer | $68.84_{\pm 0.90}$ | $69.20_{\pm 0.99}$ | $69.22_{\pm 0.43}$ | $69.40_{\pm 0.12}$ | $69.55_{\pm 0.31}$ | 0.71 |
| WikiCS | $76.26_{\pm 0.06}$ | $76.62_{\pm 0.20}$ | $76.82_{\pm 0.43}$ | $76.31_{\pm 0.40}$ | $76.26_{\pm 0.31}$ | 0.56 |
| Instagram | $59.80_{\pm 0.41}$ | $60.13_{\pm 0.84}$ | $59.88_{\pm 0.24}$ | $60.30_{\pm 0.82}$ | $60.18_{\pm 0.34}$ | 0.50 |
| Reddit | $58.87_{\pm 0.38}$ | $58.91_{\pm 0.43}$ | $58.59_{\pm 0.44}$ | $57.55_{\pm 0.30}$ | $58.14_{\pm 0.97}$ | 1.36 |

Across all five datasets, independent LLMs converge on the same per-class concept space: the choice of LLM changes the *wording* of the concepts (and, on social graphs, the granularity of individual concepts) but not the class-level concept geometry that **GCB** relies on. **GCB**'s bottleneck is therefore a property of the task and data rather than of a particular language model—on academic and social graphs alike—so its discovered concepts, explanations, and downstream predictive accuracy (Table 4) do not hinge on a single LLM choice.

## 5 Further Discussion

*GCB vs. LLM-as-predictor methods.* While some LLM-as-predictor/explainer approaches Wang et al. (2024); Chen et al. (2024a); Tang et al. (2024); Pan et al. (2025) can produce predictions accompanied by natural language explanations that may appear informative, they are fundamentally different. (1) Their explanations are inherently *post-hoc*: the generated text is not guaranteed to faithfully reflect the actual reasoning process, and how these explanations are produced remains another black box. In contrast, **GCB** makes predictions directly based on the semantics of concepts, ensuring that explanations are faithful and aligned with the model's prediction. (2) **GCB** requires access to LLMs only during training. At inference time, no LLM queries are needed. In comparison, LLM-as-predictor methods rely on querying LLMs for each prediction, which incurs substantial computational and monetary costs.

*GCB's generalizability across different graph types.* The performance of **GCB** largely depends on the quality of the proposed concept space and the effectiveness of the graph-concept alignment model—both of which rely on LLMs for semantic understanding and reasoning over graph instances. To date, LLM-based graph reasoning has primarily focused on text-attributed graphs, which motivates our choice of such graphs as the starting point for exploring **GCB**. Nevertheless, **GCB** holds strong potential for broader applicability to diverse graph types, such as molecular and biomedical graphs, provided that suitable LLM-driven interfaces Wang et al. (2025); Lee et al. (2025a); Bran et al. (2023) are available to bridge domain-specific graph structures with high-level concepts. We plan to explore this direction as future work.

## 6 Conclusion

We present **GCB** as a novel solution for self-explainable text-attributed graph learning. **GCB** maps graph inputs into a human-interpretable concept space, where each concept is in natural language and carries

semantics. Predictions are then made directly based on these concepts. We conduct extensive experiments and case studies on five real-world datasets from diverse domains to demonstrate the effectiveness of **GCB** both as a predictor and as an explainer.

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

# Appendix

# A  Additional Related work

## A.1  Concept Bottleneck Model

Concept Bottleneck Models (CBMs) aim to improve model transparency by first mapping inputs into an interpretable set of human-defined concepts (the concept bottleneck), and then making predictions based on those concepts. The original CBM framework Koh et al. (2020) is trained on datasets where each input is annotated with both class labels and corresponding concept labels. At test time, the model predicts concepts from the input and uses them as intermediate representations to produce the final output via a classifier or regressor. This process enhances interpretability and enables human intervention by allowing concept-level edits. However, original CBM Koh et al. (2020) requires substantial human effort to define the concept space and annotate each training sample with concept labels, which can be both time-consuming and labor-intensive. Moreover, they often suffer from suboptimal predictive performance. To address these limitations, Yuksekgonul et al. Yuksekgonul et al. (2023) propose a post-hoc CBM that converts any pretrained model into a concept bottleneck model. Their approach leverages multimodal approaches such as CLIP Radford et al. (2021) to align the input space (e.g., images) with a concept space (e.g., text), thereby reducing the need for explicitly labeled concept data. Nevertheless, this method still requires human expertise or additional learning steps to define the concept subspace. In a concurrent work, Oikarinen et al. Oikarinen et al. (2023) build upon similar ideas but go further by proposing a label-free CBM. They also utilize CLIP's image and text encoders to map inputs to concepts, while fully automating the construction of the concept space using large language models (LLMs). Both approaches Oikarinen et al. (2023); Yuksekgonul et al. (2023) report maintaining competitive predictive performance while improving interpretability.

In addition to these, several works explore specific challenges and extended settings of CBMs. Shang et al. Shang et al. (2024) address the concept completeness problem by proposing to recover missing concepts through transforming complemented vectors with unclear semantics into potential concepts. Shin et al. Shin et al. (2023) conduct in-depth analyses of intervention strategies in CBMs; for instance, they investigate which concept selection criteria are most cost-efficient yet effective in improving task performance. Kim et al. Kim et al. (2023) propose a probabilistic Concept Bottleneck Model to tackle ambiguity in concept prediction, which can undermine model reliability. Their approach explicitly models uncertainty in the concept space and provides explanations incorporating both the predicted concepts and their associated uncertainties. It is also worth mentioning that Xu et al. Xu et al. (2025) introduce a Graph Concept Bottleneck Model that facilitates the modeling of concept relationships by constructing a graph of latent concepts. Although it shares a similar name with our model, it tackles fundamentally different challenges. All of the aforementioned works focus on Euclidean input spaces such as images, and how to adapt Concept Bottleneck Models to graph data remains largely unexplored.

# B  Additional details on methodology

## B.1  Prompt details

> **Prompt for self-supervised concept annotations**
>
> ```
> Given {graphML} and {dataset-details}.
>     1. Provide summary and context analysis on the graph.
>     2. Identify a list of key concepts and themes presented in the graph.
>
> GraphML refers to the graph markup language Brandes et al. (2000) used for describing
> the graph (ego-net).  We sample up to 20 neighboring nodes to control the prompt length.
> dataset-details provides a detailed description of the graph dataset, including what each
> node/edge represents and relevant contextual information.
> ```

**Prompt for Global Concept Proposal**

```
In the domain of {dataset-domain}, list the related concepts/keywords for classifying the item
as {category}.

Dataset-domain briefly describes the dataset's domain or context, and category is the name
of a class label from the downstream classification task.  We apply this prompt to each class
label and aggregate the generated concepts to form the initial concept pool.
```

**Prompt for Instance-Based Concept Extractionl**

```
Given a {graphML} and {dataset-details}.
    1. Provide summary and context analysis on the graph.
    2. Identify a list of key concepts presented in the graph that are most important
       for determining its classification within the {dataset-domain}, which includes the
       following categories:  {category-list}.

GraphML refers to the graph markup language used for describing the graph (or ego-net if the
instance is a node).  dataset-details provides a detailed description of the graph dataset,
dataset-domain briefly describes the dataset's domain or context.  category-list is the
complete list of categories for the classification task to guide the LLM toward generating
concepts that are helpful in predicting class labels.  Only the outputted concept list from
the second step is collected.
```

### B.2    Detailed procedures

**Instance-Based Concept Extraction**. We sample $m$ graph instances from each class and apply the prompt to each sampled graph instance, resulting in a large set of candidate concepts. We then identify a subset of concepts that are highly relevant to each class, distinct from those used by other classes, and useful for improving class discrimination. Specifically, for each class $y$, we calculate the class-wise concept activation score as:

$$\bar{C}_y = \frac{1}{|\mathcal{D}_y|} \sum_{x_i \in \mathcal{D}_y} C_i, \tag{7}$$

where $\mathcal{D}_y$ denotes the set of instances belonging to class $y$, and $C_i$ is the concept activation vector for instance $x_i$. Each element $C_i^{(j)}$ represents the activation score (e.g., cosine similarity) between the instance representation $f_\theta^{\text{GNN}}(x_i)$ and the embedding of the $j$-th concept $f^{\text{LM}}(c_j)$.

We then compute the discriminative score of concept $j$ for class $y$ as:

$$\text{score}_j(y) = \bar{C}_y^{(j)} - \frac{1}{|\mathcal{Y}| - 1} \sum_{y' \neq y} \bar{C}_{y'}^{(j)}, \tag{8}$$

where $\mathcal{Y}$ is the set of all class labels, and $\bar{C}_y^{(j)}$ denotes the average activation of concept $j$ for class $y$.

Finally, for each class, we select the top-$k$ concepts with the highest discriminative scores:

$$\mathcal{C}^{\text{inst}} = \text{Top-}k_j(\text{score}_j(y)). \tag{9}$$

**Details of Concept Filtering Process.**   Following similar procedures to Oikarinen et al. (2023), we apply a post-processing pipeline to refine the set of candidate concepts. The pipeline consists of the following steps:

*(1) Removing overly long concepts.* Long concepts may reduce both interpretability and generalizability. We therefore tokenize each concept and discard those containing more than 10 tokens.

*(2) Removing concepts overly similar to class labels.* Concepts that are identical or highly similar to class labels undermine the purpose of explanation. To mitigate this issue, we compute the cosine similarity between

Table 5: Summary statistics of source and target datasets.

| Dataset | #Nodes | #Edges | Type | Domain | #Class |
|---|---|---|---|---|---|
| Ele-Computers | 87,229 | 721,081 | Co-purchase | E-commerce | 10 |
| PubMed | 19,717 | 44,338 | Citation | Biomedicine | 3 |
| Books-History | 41,551 | 358,574 | Co-purchase | E-commerce | 12 |
| Books-Children | 76,875 | 1,554,578 | Co-purchase | E-commerce | 24 |
| Sports-Fitness | 173,055 | 1,773,500 | Co-purchase | E-commerce | 13 |
| Cora | 2,708 | 5,429 | Citation | Computer Science | 7 |
| CiteSeer | 3,186 | 4,277 | Citation | Computer Science | 6 |
| Instagram | 11,339 | 144,010 | User-Post | Social Media | 2 |
| Reddit | 33,434 | 198,448 | Post-Comment | Social Media | 2 |
| WikiCS | 11,701 | 215,863 | Article Link | Wikipedia | 10 |

the Sentence-BERT embeddings of each concept and each class label, and filter out any concept with a similarity score greater than 0.85.

*(3) Removing redundant concepts.* To reduce redundancy, we compute pairwise cosine similarity among concepts and remove any concept whose similarity with a retained concept exceeds 0.85.

## C Supplemental experiment setups

### C.1 Details of the datasets

In this section, we summarize the basic statistics of the datasets in our experimental evaluation in Table 5. All datasets used in our study are publicly available and come from diverse domains, including social media networks, citation graphs, e-commerce graphs, and Wikipedia article networks. Each node is associated with a class label, and most datasets contain more than two classes. The class distributions are imbalanced in these datasets. Therefore, in our experiments, we report node classification performance using the (Macro-)F1 score and balanced accuracy (BACC). For all datasets and settings, we adopt a default train/validation/test split of 20%/20%/50%, with the remain- ing 10% held out to investigate the effect of increased training data in later experiments. We use an inductive splitting, where test nodes are entirely unseen during training.

### C.2 Implementation details

For all GNN-based methods, including those that use GNNs as backbones, we set the hidden dimension to 64 and the number of GNN layers to 2. For GAT and GT models, we use 4 attention heads. For the Set2set contrastive learning, we set the number of augmented views as 10, and temperature $\tau$ as 0.07. We ask the LLM to generate 10 concepts per instance and keep the top-100 most relevant ones to form the candidate concept pool $\mathcal{C}^{\text{candidate}}$; the information bottleneck then selects the final $K = 30$ concepts from this pool (Section 3.3). We use the pretrained Sentence-BERT Reimers & Gurevych (2019) model as the text encoder for all baselines. For SEGNN Dai & Wang (2021), the original implementation requires access to all training nodes at test time in order to identify the closest neighbors and make predictions based on their labels. However, this approach is incompatible with our inductive setting, where the model is not permitted to access training instances during inference. To address this, we modify the implementation by introducing a small memory buffer that stores 5 randomly selected nodes per class from the training set. During testing, the model is restricted to retrieving neighbors only from this buffer. For all self-explainable graph learning baselines, we follow the default hyperparameter settings provided in their open-source implementations. All experiments are conducted on four NVIDIA L40S GPUs. We access GPT-3.5 via the OpenAI API and set the temperature to 0 during graph summary and concept generation to avoid randomness.

## D Complexity analysis

The primary overhead of **GCB** lies in the pretraining stage, where a graph encoder is aligned with a semantically meaningful concept space using large language models (LLMs). However, this pretraining is performed once and can be reused across downstream datasets without incurring additional cost. During

Table 6: Node classification performance in *regular settings*. The best-performing interpretable GNN on each dataset is highlighted in grey, and the overall best-performing method is **bolded**.

| Method | Cora | | Citeseer | | Instagram | | Reddit | | WikiCS | |
|---|---|---|---|---|---|---|---|---|---|---|
| | F1 (%) | BACC (%) | F1 (%) | BACC (%) | F1 (%) | BACC (%) | F1 (%) | BACC (%) | F1 (%) | BACC (%) |
| MLP | $68.00_{(0.89)}$ | $67.49_{(0.76)}$ | $63.81_{(0.37)}$ | $64.29_{(0.34)}$ | $53.78_{(0.59)}$ | $53.76_{(0.58)}$ | $53.34_{(0.77)}$ | $53.39_{(0.76)}$ | $69.22_{(0.51)}$ | $69.25_{(0.57)}$ |
| GCN | $72.38_{(0.58)}$ | $71.95_{(0.49)}$ | $62.47_{(0.29)}$ | $63.05_{(0.33)}$ | $53.63_{(0.84)}$ | $53.62_{(0.82)}$ | $54.49_{(1.19)}$ | $54.64_{(1.03)}$ | $67.45_{(1.76)}$ | $68.84_{(1.82)}$ |
| GAT | $\mathbf{74.58}_{(0.95)}$ | $\mathbf{74.42}_{(0.90)}$ | $63.92_{(0.92)}$ | $64.47_{(0.91)}$ | $55.71_{(1.06)}$ | $55.74_{(1.10)}$ | $\mathbf{56.29}_{(0.60)}$ | $56.30_{(0.59)}$ | $67.54_{(1.68)}$ | $68.14_{(1.59)}$ |
| SAGE | $70.59_{(0.68)}$ | $70.66_{(0.80)}$ | $\mathbf{65.09}_{(0.62)}$ | $\mathbf{65.52}_{(0.64)}$ | $54.49_{(0.46)}$ | $54.50_{(0.46)}$ | $55.33_{(0.38)}$ | $55.33_{(0.38)}$ | $\mathbf{72.96}_{(0.30)}$ | $\mathbf{72.74}_{(0.40)}$ |
| GT | $72.36_{(1.96)}$ | $72.18_{(1.56)}$ | $64.40_{(0.76)}$ | $64.93_{(0.68)}$ | $54.79_{(0.40)}$ | $54.77_{(0.39)}$ | $56.15_{(0.39)}$ | $56.15_{(0.39)}$ | $72.27_{(0.52)}$ | $72.46_{(0.56)}$ |
| DIR-GNN | $73.03_{(2.62)}$ | $72.51_{(1.90)}$ | $62.10_{(0.58)}$ | $64.67_{(0.50)}$ | $\mathbf{56.76}_{(1.24)}$ | $\mathbf{57.37}_{(0.95)}$ | $55.34_{(1.81)}$ | $\mathbf{57.18}_{(0.48)}$ | $67.14_{(3.60)}$ | $66.26_{(3.83)}$ |
| GIB | $66.81_{(4.23)}$ | $67.23_{(4.02)}$ | $49.28_{(14.03)}$ | $53.88_{(11.42)}$ | $40.72_{(8.44)}$ | $51.52_{(1.86)}$ | $38.84_{(8.18)}$ | $51.49_{(2.11)}$ | $45.30_{(18.50)}$ | $45.38_{(14.85)}$ |
| VGIB | $63.46_{(28.19)}$ | $64.59_{(25.11)}$ | $53.90_{(19.24)}$ | $56.99_{(16.88)}$ | $39.64_{(1.64)}$ | $50.29_{(0.58)}$ | $33.68_{(1.13)}$ | $50.12_{(0.22)}$ | $61.44_{(25.27)}$ | $62.90_{(22.46)}$ |
| SEGNN | $49.90_{(4.09)}$ | $53.07_{(3.30)}$ | $52.12_{(5.51)}$ | $55.67_{(4.11)}$ | $44.71_{(2.56)}$ | $51.04_{(0.49)}$ | $53.53_{(1.66)}$ | $54.59_{(0.90)}$ | $28.87_{(3.57)}$ | $34.71_{(2.78)}$ |
| **GCB** | $70.54_{(1.33)}$ | $71.41_{(0.88)}$ | $63.22_{(0.50)}$ | $63.54_{(0.49)}$ | $\mathbf{56.76}_{(0.55)}$ | $56.71_{(0.51)}$ | $55.06_{(0.72)}$ | $55.11_{(0.72)}$ | $68.82_{(0.41)}$ | $70.64_{(0.82)}$ |

the main training phase, where we optimize the information bottleneck criteria, the dominant cost comes from computing the gate vector $g$ (via a lightweight MLP) and training the classifier $\text{MLP}^{\text{cls}}$ on the masked concept representations. This results in a per-step complexity of $O(BKH)$, where $B$ is the batch size, $K$ is the number of candidate concepts, and $H$ is the hidden dimension of the MLP. The final predictor, after concept selection, operates on a reduced concept set and is simply a small MLP, which is highly efficient in both training and inference. At inference time, **GCB** consists of a frozen graph encoder (e.g., a GNN) followed by a fixed MLP classifier over selected concepts, making its runtime complexity comparable to that of a standard GNN model.

# E  Additional results

## E.1  Robustness Evaluation

We report the results for all additional upsampling ratios $\gamma \in \{2, 3, 10\}$ in Table 7, Table 8, and Table 9, respectively. Results for different perturbation ratios $\rho \in \{0.05, 0.1, 0.2, 0.5\}$ are shown in Table 10, Table 11, Table 12, and Table 13.

We emphasize that the test splits used for different upsampling ratios are not aligned, making direct comparison across these settings inappropriate. While a larger upsampling ratio increases the distribution shift between the training and test sets, it may also lead to a more balanced class distribution in the training or test data, which can sometimes improve test performance. Regarding the perturbation setting, we observe that **GCB** is the least affected by structural perturbation. We attribute this to the use of a fixed pretrained encoder, which is not updated during task-specific training. As a result, perturbing the training graph does not alter the graph embedding function. Moreover, the data augmentation used during pretraining also contributes to **GCB**'s robustness under structural noise. Interestingly, across all baseline methods, we do not observe a consistent trend correlating performance with increasing perturbation ratio. One possible explanation is that, for perturbation-sensitive models, even a small perturbation (e.g., $\rho = 0.05$) significantly degrades performance, and the marginal impact of further perturbation is limited. Furthermore, recent studies such as Han et al. (2023) have shown that some GNNs can perform well even when trained without graph structure—effectively functioning like MLPs—and still generalize well when tested with full graph connectivity. When the perturbation ratio is large, models may similarly learn to disregard noisy structure, exhibiting behavior consistent with such MLP-based approaches and mitigating the negative effects of edge pertubation.

Table 7: Node classification performance in *OOD settings* with upsampling ratio $\gamma = 2$. The best-performing interpretable GNN on each dataset is highlighted in grey, and the overall best-performing method is **bolded**.

| Method | Cora | | Citeseer | | Instagram | | Reddit | | WikiCS | |
|---|---|---|---|---|---|---|---|---|---|---|
| | F1 (%) | BACC (%) | F1 (%) | BACC (%) | F1 (%) | BACC (%) | F1 (%) | BACC (%) | F1 (%) | BACC (%) |
| MLP | $46.52_{(0.59)}$ | $57.50_{(0.69)}$ | $44.89_{(0.70)}$ | $60.49_{(0.77)}$ | $35.55_{(0.59)}$ | $51.54_{(0.20)}$ | $17.03_{(0.51)}$ | $51.28_{(0.56)}$ | $53.82_{(0.42)}$ | $63.23_{(0.40)}$ |
| GCN | $\mathbf{56.10}_{(0.46)}$ | $\mathbf{64.04}_{(0.52)}$ | $41.06_{(0.43)}$ | $56.08_{(0.57)}$ | $39.36_{(3.25)}$ | $52.54_{(0.86)}$ | $16.65_{(0.73)}$ | $50.74_{(0.20)}$ | $53.80_{(0.55)}$ | $60.37_{(1.37)}$ |
| GAT | $52.54_{(1.48)}$ | $63.29_{(1.31)}$ | $44.71_{(0.68)}$ | $60.63_{(0.56)}$ | $33.42_{(0.37)}$ | $51.49_{(0.12)}$ | $13.06_{(0.28)}$ | $49.78_{(0.39)}$ | $\mathbf{56.41}_{(2.24)}$ | $64.32_{(1.67)}$ |
| SAGE | $40.39_{(1.19)}$ | $50.69_{(1.08)}$ | $40.99_{(0.99)}$ | $56.02_{(0.78)}$ | $35.71_{(0.42)}$ | $51.76_{(0.31)}$ | $15.97_{(0.35)}$ | $50.74_{(0.51)}$ | $49.65_{(0.64)}$ | $60.20_{(0.83)}$ |
| GT | $42.83_{(1.35)}$ | $51.62_{(1.59)}$ | $40.57_{(1.22)}$ | $56.93_{(0.90)}$ | $33.83_{(0.42)}$ | $51.47_{(0.28)}$ | $13.22_{(0.33)}$ | $50.14_{(0.34)}$ | $51.10_{(0.69)}$ | $59.51_{(0.99)}$ |
| DIR-GNN | $18.54_{(2.90)}$ | $40.48_{(2.24)}$ | $15.18_{(0.70)}$ | $42.44_{(0.52)}$ | $26.74_{(0.00)}$ | $50.00_{(0.00)}$ | $8.46_{(0.00)}$ | $50.00_{(0.00)}$ | $23.41_{(1.92)}$ | $42.89_{(0.55)}$ |
| GIB | $21.45_{(3.55)}$ | $40.93_{(1.77)}$ | $16.98_{(4.91)}$ | $43.81_{(3.24)}$ | $26.76_{(0.02)}$ | $50.01_{(0.01)}$ | $8.53_{(0.06)}$ | $49.99_{(0.06)}$ | $23.93_{(1.12)}$ | $41.10_{(1.63)}$ |
| VGIB | $45.60_{(4.00)}$ | $58.19_{(2.70)}$ | $15.61_{(1.75)}$ | $44.07_{(1.00)}$ | $26.74_{(0.00)}$ | $50.00_{(0.00)}$ | $8.46_{(0.00)}$ | $50.00_{(0.00)}$ | $54.31_{(0.90)}$ | $63.54_{(0.66)}$ |
| SEGNN | $40.04_{(2.47)}$ | $51.44_{(2.36)}$ | $25.59_{(2.76)}$ | $45.69_{(1.58)}$ | $26.74_{(0.00)}$ | $50.00_{(0.00)}$ | $8.46_{(0.00)}$ | $50.00_{(0.00)}$ | $37.26_{(1.18)}$ | $49.80_{(0.91)}$ |
| **GCB** | $54.23_{(0.00)}$ | $62.97_{(1.15)}$ | $\mathbf{57.46}_{(0.85)}$ | $\mathbf{65.48}_{(0.65)}$ | $\mathbf{53.20}_{(0.81)}$ | $\mathbf{55.89}_{(0.82)}$ | $\mathbf{43.74}_{(0.77)}$ | $\mathbf{57.99}_{(0.71)}$ | $55.19_{(0.72)}$ | $\mathbf{66.36}_{(0.62)}$ |

Table 8: Node classification performance in *OOD settings* with upsampling ratio $\gamma = 3$. The best-performing interpretable GNN on each dataset is highlighted in grey, and the overall best-performing method is **bolded**.

| Method | Cora | | Citeseer | | Instagram | | Reddit | | WikiCS | |
|---|---|---|---|---|---|---|---|---|---|---|
| | F1 (%) | BACC (%) | F1 (%) | BACC (%) | F1 (%) | BACC (%) | F1 (%) | BACC (%) | F1 (%) | BACC (%) |
| MLP | $43.78_{(0.52)}$ | $54.85_{(0.57)}$ | $45.73_{(0.59)}$ | $58.72_{(0.71)}$ | $37.04_{(0.67)}$ | $52.33_{(0.25)}$ | $16.37_{(0.61)}$ | $51.37_{(0.32)}$ | $55.01_{(0.33)}$ | $65.33_{(1.15)}$ |
| GCN | $\mathbf{57.28}_{(1.35)}$ | $\mathbf{67.86}_{(0.98)}$ | $41.59_{(0.68)}$ | $56.90_{(0.80)}$ | $37.73_{(3.06)}$ | $51.58_{(0.59)}$ | $16.00_{(0.35)}$ | $49.23_{(0.48)}$ | $54.99_{(1.28)}$ | $61.65_{(2.33)}$ |
| GAT | $52.81_{(1.43)}$ | $60.66_{(1.28)}$ | $43.09_{(1.45)}$ | $58.16_{(1.39)}$ | $34.77_{(1.08)}$ | $51.50_{(0.38)}$ | $13.85_{(0.44)}$ | $49.44_{(0.28)}$ | $\mathbf{56.99}_{(0.10)}$ | $65.85_{(0.84)}$ |
| SAGE | $51.40_{(1.77)}$ | $62.46_{(1.54)}$ | $36.25_{(1.39)}$ | $52.63_{(1.14)}$ | $35.04_{(0.55)}$ | $51.47_{(0.27)}$ | $14.36_{(0.17)}$ | $48.96_{(0.40)}$ | $52.43_{(0.78)}$ | $60.92_{(0.97)}$ |
| GT | $47.70_{(1.31)}$ | $58.26_{(1.19)}$ | $36.12_{(1.62)}$ | $53.10_{(1.23)}$ | $33.16_{(0.47)}$ | $50.15_{(0.37)}$ | $14.18_{(0.15)}$ | $49.59_{(0.21)}$ | $54.83_{(0.89)}$ | $62.58_{(1.07)}$ |
| DIR-GNN | $20.13_{(2.75)}$ | $41.89_{(1.50)}$ | $14.70_{(0.34)}$ | $43.15_{(0.38)}$ | $26.74_{(0.00)}$ | $50.00_{(0.00)}$ | $8.46_{(0.00)}$ | $50.00_{(0.00)}$ | $23.60_{(1.40)}$ | $42.84_{(0.57)}$ |
| GIB | $22.43_{(5.91)}$ | $41.53_{(4.24)}$ | $17.72_{(5.88)}$ | $44.14_{(4.31)}$ | $26.74_{(0.00)}$ | $50.01_{(0.02)}$ | $8.48_{(0.04)}$ | $50.01_{(0.02)}$ | $20.30_{(7.70)}$ | $35.16_{(9.59)}$ |
| VGIB | $44.05_{(2.53)}$ | $57.14_{(1.92)}$ | $17.14_{(4.35)}$ | $44.50_{(2.44)}$ | $26.74_{(0.00)}$ | $50.00_{(0.00)}$ | $8.46_{(0.00)}$ | $50.00_{(0.00)}$ | $54.74_{(1.32)}$ | $63.15_{(1.20)}$ |
| SEGNN | $29.92_{(1.05)}$ | $46.62_{(0.81)}$ | $25.15_{(9.17)}$ | $43.70_{(6.98)}$ | $26.74_{(0.00)}$ | $50.00_{(0.00)}$ | $8.46_{(0.00)}$ | $50.00_{(0.00)}$ | $27.85_{(1.02)}$ | $43.33_{(1.50)}$ |
| **GCB** | $54.99_{(0.00)}$ | $65.00_{(0.00)}$ | $\mathbf{57.85}_{(0.27)}$ | $\mathbf{65.62}_{(0.79)}$ | $\mathbf{54.54}_{(0.17)}$ | $\mathbf{56.39}_{(0.36)}$ | $\mathbf{45.82}_{(0.31)}$ | $\mathbf{60.65}_{(0.83)}$ | $54.97_{(0.37)}$ | $\mathbf{66.70}_{(0.40)}$ |

Table 9: Node classification performance in *OOD settings* with upsampling ratio $\gamma = 10$. The best-performing interpretable GNN on each dataset is highlighted in grey, and the overall best-performing method is **bolded**.

| Method | Cora | | Citeseer | | Instagram | | Reddit | | WikiCS | |
|---|---|---|---|---|---|---|---|---|---|---|
| | F1 (%) | BACC (%) | F1 (%) | BACC (%) | F1 (%) | BACC (%) | F1 (%) | BACC (%) | F1 (%) | BACC (%) |
| MLP | $47.58_{(0.44)}$ | $59.14_{(0.61)}$ | $41.44_{(0.42)}$ | $56.87_{(0.70)}$ | $35.38_{(0.66)}$ | $51.29_{(0.43)}$ | $16.08_{(0.42)}$ | $50.69_{(0.34)}$ | $52.72_{(0.50)}$ | $62.79_{(0.50)}$ |
| GCN | $\mathbf{62.08}_{(1.59)}$ | $69.26_{(1.83)}$ | $43.58_{(0.45)}$ | $54.79_{(0.34)}$ | $47.15_{(4.10)}$ | $55.23_{(1.48)}$ | $17.35_{(0.97)}$ | $49.91_{(0.17)}$ | $62.47_{(1.15)}$ | $64.89_{(1.38)}$ |
| GAT | $60.32_{(1.56)}$ | $68.94_{(1.15)}$ | $48.46_{(0.66)}$ | $61.74_{(0.80)}$ | $35.80_{(0.67)}$ | $51.70_{(0.30)}$ | $15.35_{(1.35)}$ | $51.40_{(0.38)}$ | $57.17_{(1.59)}$ | $60.82_{(1.69)}$ |
| SAGE | $50.49_{(1.06)}$ | $57.96_{(1.12)}$ | $35.75_{(1.49)}$ | $53.22_{(0.90)}$ | $37.94_{(0.94)}$ | $52.14_{(0.40)}$ | $16.70_{(0.73)}$ | $52.08_{(0.22)}$ | $\mathbf{62.88}_{(0.18)}$ | $\mathbf{72.40}_{(1.02)}$ |
| GT | $47.31_{(2.61)}$ | $56.27_{(1.92)}$ | $30.80_{(1.22)}$ | $50.68_{(0.82)}$ | $37.51_{(0.46)}$ | $52.90_{(0.16)}$ | $17.33_{(0.79)}$ | $51.58_{(0.24)}$ | $62.18_{(0.80)}$ | $68.79_{(1.87)}$ |
| DIR-GNN | $22.04_{(3.39)}$ | $42.60_{(2.67)}$ | $15.56_{(1.05)}$ | $42.93_{(0.37)}$ | $26.74_{(0.00)}$ | $50.00_{(0.00)}$ | $8.46_{(0.00)}$ | $50.00_{(0.00)}$ | $23.89_{(0.58)}$ | $42.04_{(0.65)}$ |
| GIB | $26.30_{(8.89)}$ | $44.06_{(5.96)}$ | $14.94_{(0.54)}$ | $42.42_{(0.68)}$ | $26.92_{(0.31)}$ | $50.06_{(0.13)}$ | $8.46_{(0.02)}$ | $49.98_{(0.05)}$ | $25.39_{(2.25)}$ | $38.19_{(1.06)}$ |
| VGIB | $60.87_{(3.20)}$ | $69.42_{(3.04)}$ | $24.29_{(6.80)}$ | $48.20_{(3.96)}$ | $26.74_{(0.00)}$ | $50.00_{(0.00)}$ | $8.46_{(0.00)}$ | $50.00_{(0.00)}$ | $61.85_{(1.81)}$ | $69.02_{(1.35)}$ |
| **GCB** | $61.68_{(1.63)}$ | $\mathbf{69.55}_{(1.11)}$ | $\mathbf{58.08}_{(0.34)}$ | $\mathbf{65.52}_{(0.25)}$ | $\mathbf{52.17}_{(2.34)}$ | $\mathbf{55.57}_{(1.15)}$ | $\mathbf{44.77}_{(1.34)}$ | $\mathbf{55.75}_{(0.43)}$ | $60.66_{(0.97)}$ | $71.22_{(1.43)}$ |

Table 10: Node classification performance in *adversarial settings* with perturbation ratio $\rho = 0.05$. The best-performing interpretable GNN on each dataset is highlighted in grey, and the overall best-performing method is **bolded**.

| Method | Cora | | Citeseer | | Instagram | | Reddit | | WikiCS | |
|---|---|---|---|---|---|---|---|---|---|---|
| | F1 (%) | BACC (%) | F1 (%) | BACC (%) | F1 (%) | BACC (%) | F1 (%) | BACC (%) | F1 (%) | BACC (%) |
| MLP | $46.79_{(0.88)}$ | $58.83_{(0.84)}$ | $38.16_{(0.41)}$ | $53.71_{(0.37)}$ | $37.69_{(0.44)}$ | $52.53_{(0.15)}$ | $16.19_{(0.46)}$ | $51.58_{(0.44)}$ | $53.96_{(0.34)}$ | $64.83_{(0.25)}$ |
| GCN | $58.93_{(1.32)}$ | $67.16_{(1.38)}$ | $46.96_{(0.95)}$ | $58.48_{(1.13)}$ | $42.47_{(4.66)}$ | $52.11_{(1.09)}$ | $15.95_{(0.86)}$ | $50.46_{(0.68)}$ | $62.44_{(0.37)}$ | $68.39_{(0.49)}$ |
| GAT | $55.01_{(1.93)}$ | $61.57_{(1.94)}$ | $43.59_{(1.57)}$ | $57.78_{(1.21)}$ | $34.32_{(1.37)}$ | $51.16_{(0.53)}$ | $17.03_{(0.95)}$ | $51.28_{(0.42)}$ | $58.93_{(2.81)}$ | $66.43_{(2.35)}$ |
| SAGE | $53.45_{(2.23)}$ | $57.21_{(2.34)}$ | $42.45_{(1.44)}$ | $57.74_{(0.95)}$ | $40.93_{(6.08)}$ | $52.32_{(0.67)}$ | $16.00_{(0.61)}$ | $51.42_{(0.61)}$ | $61.58_{(0.22)}$ | $68.96_{(1.30)}$ |
| GT | $38.25_{(2.04)}$ | $45.19_{(1.18)}$ | $39.80_{(3.32)}$ | $55.30_{(2.16)}$ | $34.43_{(1.15)}$ | $51.44_{(0.30)}$ | $14.35_{(0.74)}$ | $51.02_{(0.27)}$ | $56.30_{(1.16)}$ | $64.35_{(1.49)}$ |
| DIR-GNN | $\mathbf{73.48}_{(1.08)}$ | $\mathbf{72.72}_{(1.37)}$ | $62.03_{(0.64)}$ | $\mathbf{64.60}_{(0.54)}$ | $55.78_{(2.54)}$ | $56.70_{(1.42)}$ | $54.64_{(2.70)}$ | $57.12_{(1.00)}$ | $65.05_{(1.45)}$ | $63.77_{(1.50)}$ |
| GIB | $58.60_{(15.18)}$ | $59.17_{(14.38)}$ | $45.60_{(17.26)}$ | $50.91_{(13.49)}$ | $40.96_{(8.68)}$ | $51.59_{(1.93)}$ | $38.57_{(7.79)}$ | $51.70_{(2.56)}$ | $40.07_{(16.67)}$ | $40.14_{(12.96)}$ |
| VGIB | $21.17_{(26.63)}$ | $26.65_{(23.68)}$ | $53.90_{(19.09)}$ | $57.17_{(16.80)}$ | $38.99_{(0.34)}$ | $50.07_{(0.14)}$ | $34.58_{(2.56)}$ | $50.29_{(0.47)}$ | $\mathbf{72.78}_{(1.07)}$ | $\mathbf{72.45}_{(1.37)}$ |
| SEGNN | $55.79_{(1.48)}$ | $59.39_{(1.03)}$ | $60.06_{(0.69)}$ | $62.95_{(0.74)}$ | $54.75_{(1.01)}$ | $55.22_{(0.91)}$ | $55.58_{(0.36)}$ | $\mathbf{55.99}_{(0.30)}$ | $37.35_{(0.71)}$ | $41.85_{(1.12)}$ |
| **GCB** | $70.75_{(0.85)}$ | $71.34_{(1.05)}$ | $\mathbf{63.20}_{(0.76)}$ | $63.52_{(0.78)}$ | $\mathbf{56.79}_{(0.60)}$ | $\mathbf{56.72}_{(0.59)}$ | $54.93_{(0.78)}$ | $54.98_{(0.79)}$ | $68.70_{(0.42)}$ | $70.60_{(0.46)}$ |

Table 11: Node classification performance in *adversarial settings* with perturbation ratio $\rho = 0.1$. The best-performing interpretable GNN on each dataset is highlighted in grey, and the overall best-performing method is **bolded**.

| Method | Cora | | Citeseer | | Instagram | | Reddit | | WikiCS | |
|---|---|---|---|---|---|---|---|---|---|---|
| | F1 (%) | BACC (%) | F1 (%) | BACC (%) | F1 (%) | BACC (%) | F1 (%) | BACC (%) | F1 (%) | BACC (%) |
| MLP | $45.04_{(1.20)}$ | $57.94_{(0.91)}$ | $41.94_{(0.33)}$ | $57.38_{(0.21)}$ | $34.65_{(0.60)}$ | $51.57_{(0.30)}$ | $18.09_{(0.43)}$ | $51.09_{(0.58)}$ | $54.13_{(0.30)}$ | $64.72_{(0.64)}$ |
| GCN | $65.19_{(1.66)}$ | $67.62_{(1.61)}$ | $46.43_{(1.13)}$ | $58.40_{(1.17)}$ | $38.56_{(1.38)}$ | $51.96_{(0.60)}$ | $18.17_{(1.20)}$ | $50.51_{(0.67)}$ | $63.15_{(2.44)}$ | $68.98_{(2.14)}$ |
| GAT | $63.56_{(1.59)}$ | $68.88_{(1.40)}$ | $43.79_{(1.41)}$ | $58.11_{(0.90)}$ | $35.94_{(2.11)}$ | $51.83_{(0.55)}$ | $17.30_{(1.53)}$ | $51.83_{(0.72)}$ | $58.79_{(1.66)}$ | $67.83_{(1.26)}$ |
| SAGE | $47.78_{(0.92)}$ | $55.17_{(0.52)}$ | $40.60_{(0.80)}$ | $56.65_{(0.63)}$ | $38.86_{(1.28)}$ | $52.61_{(0.62)}$ | $16.93_{(0.48)}$ | $51.67_{(0.34)}$ | $59.90_{(0.67)}$ | $66.69_{(0.61)}$ |
| GT | $40.32_{(1.93)}$ | $46.77_{(1.27)}$ | $27.14_{(1.44)}$ | $46.19_{(1.04)}$ | $35.26_{(0.73)}$ | $51.95_{(0.35)}$ | $16.80_{(1.02)}$ | $51.26_{(0.49)}$ | $60.67_{(0.99)}$ | $67.86_{(1.05)}$ |
| DIR-GNN | $\mathbf{71.70}_{(2.79)}$ | $71.04_{(2.08)}$ | $61.84_{(1.36)}$ | $\mathbf{64.42}_{(1.24)}$ | $55.55_{(2.17)}$ | $56.66_{(1.28)}$ | $\mathbf{55.41}_{(1.29)}$ | $\mathbf{57.48}_{(0.53)}$ | $64.30_{(4.20)}$ | $63.15_{(4.09)}$ |
| GIB | $55.55_{(16.26)}$ | $58.38_{(11.63)}$ | $58.99_{(4.11)}$ | $61.91_{(3.36)}$ | $41.53_{(9.29)}$ | $51.81_{(2.20)}$ | $40.23_{(8.43)}$ | $52.11_{(2.82)}$ | $30.36_{(14.24)}$ | $32.54_{(12.21)}$ |
| VGIB | $22.42_{(26.34)}$ | $28.23_{(23.26)}$ | $52.91_{(22.73)}$ | $55.81_{(19.19)}$ | $38.86_{(0.07)}$ | $50.02_{(0.03)}$ | $33.92_{(1.58)}$ | $50.16_{(0.30)}$ | $59.83_{(24.76)}$ | $60.32_{(22.57)}$ |
| SEGNN | $56.89_{(0.75)}$ | $60.23_{(0.64)}$ | $59.55_{(0.62)}$ | $62.66_{(0.62)}$ | $54.67_{(0.70)}$ | $55.42_{(0.77)}$ | $55.91_{(1.85)}$ | $56.70_{(1.09)}$ | $36.78_{(1.67)}$ | $41.06_{(1.82)}$ |
| **GCB** | $70.54_{(1.54)}$ | $\mathbf{71.31}_{(2.31)}$ | $\mathbf{63.02}_{(0.40)}$ | $63.38_{(0.44)}$ | $\mathbf{56.75}_{(0.36)}$ | $\mathbf{56.70}_{(0.38)}$ | $54.91_{(0.40)}$ | $54.95_{(0.38)}$ | $\mathbf{68.80}_{(0.30)}$ | $\mathbf{70.45}_{(0.43)}$ |

Table 12: Node classification performance in *adversarial settings* with perturbation ratio $\rho = 0.2$. The best-performing interpretable GNN on each dataset is highlighted in grey, and the overall best-performing method is **bolded**.

| Method | Cora | | Citeseer | | Instagram | | Reddit | | WikiCS | |
|---|---|---|---|---|---|---|---|---|---|---|
| | F1 (%) | BACC (%) | F1 (%) | BACC (%) | F1 (%) | BACC (%) | F1 (%) | BACC (%) | F1 (%) | BACC (%) |
| MLP | $49.30_{(0.81)}$ | $59.94_{(0.97)}$ | $42.01_{(0.43)}$ | $57.97_{(0.22)}$ | $37.57_{(3.23)}$ | $51.27_{(0.41)}$ | $15.37_{(0.09)}$ | $51.51_{(0.29)}$ | $52.92_{(0.41)}$ | $61.84_{(0.70)}$ |
| GCN | $60.24_{(0.83)}$ | $68.81_{(1.01)}$ | $47.16_{(1.29)}$ | $58.67_{(1.02)}$ | $37.70_{(2.16)}$ | $51.52_{(0.53)}$ | $17.24_{(1.82)}$ | $50.60_{(0.84)}$ | $59.73_{(0.71)}$ | $62.88_{(0.35)}$ |
| GAT | $57.88_{(2.24)}$ | $64.39_{(1.85)}$ | $44.69_{(1.35)}$ | $58.83_{(1.13)}$ | $37.82_{(1.08)}$ | $52.08_{(0.46)}$ | $14.93_{(1.04)}$ | $50.50_{(0.56)}$ | $58.14_{(2.12)}$ | $62.40_{(2.18)}$ |
| SAGE | $50.26_{(1.95)}$ | $57.82_{(1.43)}$ | $29.81_{(2.27)}$ | $49.99_{(1.55)}$ | $36.98_{(0.47)}$ | $52.00_{(0.08)}$ | $17.10_{(0.40)}$ | $50.99_{(0.43)}$ | $62.87_{(0.63)}$ | $70.36_{(0.31)}$ |
| GT | $51.12_{(2.34)}$ | $56.14_{(2.39)}$ | $32.63_{(1.41)}$ | $51.08_{(0.71)}$ | $35.34_{(0.86)}$ | $51.61_{(0.50)}$ | $16.19_{(0.68)}$ | $50.84_{(0.28)}$ | $60.47_{(0.47)}$ | $66.23_{(0.77)}$ |
| DIR-GNN | $\mathbf{71.30}_{(2.36)}$ | $\mathbf{71.11}_{(1.94)}$ | $62.54_{(0.34)}$ | $\mathbf{65.12}_{(0.38)}$ | $55.77_{(2.23)}$ | $\mathbf{56.87}_{(1.32)}$ | $54.68_{(2.36)}$ | $\mathbf{56.99}_{(0.90)}$ | $61.70_{(3.35)}$ | $60.60_{(3.45)}$ |
| GIB | $37.52_{(19.90)}$ | $42.46_{(16.75)}$ | $52.91_{(12.84)}$ | $57.50_{(9.10)}$ | $40.83_{(8.60)}$ | $51.53_{(1.89)}$ | $41.45_{(9.60)}$ | $51.65_{(2.13)}$ | $24.40_{(11.79)}$ | $27.94_{(10.41)}$ |
| VGIB | $34.11_{(32.96)}$ | $38.47_{(29.40)}$ | $52.05_{(22.62)}$ | $55.80_{(19.34)}$ | $40.36_{(3.07)}$ | $50.41_{(0.81)}$ | $33.09_{(0.08)}$ | $50.00_{(0.02)}$ | $59.54_{(24.67)}$ | $61.20_{(21.58)}$ |
| SEGNN | $55.76_{(1.87)}$ | $59.44_{(1.21)}$ | $59.94_{(0.64)}$ | $62.98_{(0.54)}$ | $55.07_{(1.63)}$ | $55.27_{(1.65)}$ | $54.56_{(0.07)}$ | $55.35_{(0.37)}$ | $35.77_{(0.70)}$ | $40.40_{(0.89)}$ |
| **GCB** | $70.40_{(1.32)}$ | $71.03_{(0.60)}$ | $\mathbf{63.14}_{(0.74)}$ | $63.47_{(0.70)}$ | $\mathbf{56.81}_{(0.28)}$ | $56.76_{(0.30)}$ | $\mathbf{55.05}_{(0.44)}$ | $55.10_{(0.45)}$ | $\mathbf{68.71}_{(0.55)}$ | $\mathbf{70.64}_{(0.59)}$ |

Table 13: Node classification performance in *adversarial settings* with perturbation ratio $\rho = 0.5$. The best-performing interpretable GNN on each dataset is highlighted in grey, and the overall best-performing method is **bolded**.

| Method | Cora | | Citeseer | | Instagram | | Reddit | | WikiCS | |
|---|---|---|---|---|---|---|---|---|---|---|
| | F1 (%) | BACC (%) | F1 (%) | BACC (%) | F1 (%) | BACC (%) | F1 (%) | BACC (%) | F1 (%) | BACC (%) |
| MLP | $47.76_{(0.43)}$ | $58.52_{(0.43)}$ | $44.65_{(0.39)}$ | $58.45_{(0.41)}$ | $35.26_{(0.57)}$ | $51.72_{(0.39)}$ | $16.78_{(0.31)}$ | $50.46_{(0.59)}$ | $55.24_{(0.16)}$ | $65.36_{(1.10)}$ |
| GCN | $52.19_{(1.00)}$ | $61.01_{(0.95)}$ | $46.07_{(1.23)}$ | $56.61_{(1.12)}$ | $42.37_{(1.34)}$ | $53.07_{(0.85)}$ | $16.20_{(1.08)}$ | $50.73_{(0.48)}$ | $65.21_{(0.63)}$ | $68.76_{(1.02)}$ |
| GAT | $54.25_{(2.86)}$ | $63.55_{(1.63)}$ | $46.58_{(0.47)}$ | $60.72_{(0.81)}$ | $37.55_{(1.31)}$ | $52.63_{(0.43)}$ | $18.61_{(0.96)}$ | $51.83_{(0.37)}$ | $59.33_{(2.20)}$ | $65.94_{(1.80)}$ |
| SAGE | $44.04_{(1.28)}$ | $50.19_{(0.89)}$ | $32.53_{(2.83)}$ | $50.50_{(1.70)}$ | $36.20_{(0.84)}$ | $52.24_{(0.19)}$ | $16.29_{(0.46)}$ | $51.15_{(0.41)}$ | $62.65_{(0.42)}$ | $\mathbf{70.73}_{(0.50)}$ |
| GT | $42.25_{(1.81)}$ | $52.13_{(2.09)}$ | $31.04_{(2.55)}$ | $49.44_{(2.11)}$ | $35.67_{(0.87)}$ | $51.02_{(0.17)}$ | $13.85_{(0.83)}$ | $50.82_{(0.54)}$ | $62.95_{(1.19)}$ | $70.35_{(1.05)}$ |
| DIR-GNN | $\mathbf{71.44}_{(0.96)}$ | $69.95_{(1.33)}$ | $62.32_{(0.72)}$ | $\mathbf{64.97}_{(0.63)}$ | $52.83_{(7.05)}$ | $55.64_{(3.03)}$ | $54.60_{(2.34)}$ | $56.17_{(1.02)}$ | $53.83_{(4.77)}$ | $54.12_{(3.56)}$ |
| GIB | $25.59_{(18.05)}$ | $31.80_{(16.51)}$ | $40.56_{(16.48)}$ | $46.87_{(13.35)}$ | $38.11_{(5.95)}$ | $50.56_{(0.69)}$ | $38.93_{(8.60)}$ | $51.73_{(2.61)}$ | $17.64_{(8.56)}$ | $23.25_{(8.04)}$ |
| VGIB | $20.64_{(27.56)}$ | $26.54_{(24.51)}$ | $54.25_{(20.38)}$ | $56.74_{(17.96)}$ | $39.00_{(0.29)}$ | $50.06_{(0.13)}$ | $36.58_{(7.07)}$ | $50.60_{(1.19)}$ | $45.45_{(31.44)}$ | $47.58_{(27.67)}$ |
| SEGNN | $56.47_{(0.72)}$ | $59.51_{(0.94)}$ | $60.23_{(0.68)}$ | $63.10_{(0.72)}$ | $54.32_{(0.52)}$ | $54.61_{(0.75)}$ | $55.81_{(1.45)}$ | $\mathbf{56.36}_{(1.57)}$ | $35.41_{(0.52)}$ | $39.84_{(0.52)}$ |
| **GCB** | $70.48_{(2.31)}$ | $\mathbf{70.80}_{(1.28)}$ | $\mathbf{63.39}_{(0.37)}$ | $63.76_{(0.38)}$ | $\mathbf{56.95}_{(0.18)}$ | $\mathbf{56.91}_{(0.19)}$ | $\mathbf{55.02}_{(0.67)}$ | $55.12_{(0.68)}$ | $\mathbf{69.17}_{(0.45)}$ | $70.45_{(0.51)}$ |

