# OpenReview forum: "Exploring Concept Subspace for Self-explainable Text-Attributed Graph Learning"
_TMLR — Under review for TMLR_

### Review · Reviewer_kRey · 2026-07-20

**Summary Of Contributions:**

**Summary**: this paper introduces Graph Concept Bottleneck (GCB), a self-explainable framework for text-attributed graph (TAG) node classification. The idea is to route predictions through a natural-language concept space rather than through extracted subgraphs (which is the standard in self-explainable GNNs). The pipeline consists of three stages: (1) *Contrastive Concept-Graph Pretraining (CCGP)* which aligns a graph encoder to a concept embedding via a set-to-set InfoNCE over LLM-annotated (graph,concept) pairs; (2) *LLM-Empowered Concept Retrieval*, which proposes a candidate concept vocabulary via both class-level and instance-level LLM prompting; and (3) *Information-Constrained Concept Optimisation*, which learns a sparse gate over the candidate concepts to select a compact, discriminative concept subset for the downstream classifier. The method is evaluated on five TAG node classification datasets under regular, out-of-distribution, and adversarial settings. The authors also provide an interpretability evaluation and a cross-LLM robustness analysis.

---

**Strengths**
1. **Use of a distinct explanation modality**: Rather than incrementally improving subgraph-based self-explainers, GCB replaces the explanations with human-readable natural-language concepts. This is well motivated since, as mentioned in the paper, subgraph-importance explainers often fail to recover truly relevant structures.
2. **Thorough robustness analysis**: The distribution-shift (label-upsampling) and adversarial (edge-perturbation) analysis provides insights on how the advantage scales and offers distinct explanations with different corruption severities.
3. **Cross-LLM concept convergence analysis**: Assessing whether the discovered bottleneck is an artifact specific of GPT-3.5 is really important and should be considered more consistently across papers. The results strengthen the claim that GCB's concept space reflects task structure rather than a particular LLM's idiosyncrasies.

---

**Limitations**
1. **Claims on node classification performance**: the claim that GBC is on par with black-box GNNs is a bit of an overstatement. In fact, in Table 6, we can see that this is not true for at least two datasets. GCB trails the best black-box GNN by ~4 points on Cora and ~4 F1 points on WikiCS. Those gaps appear to exceed the reported standard deviations. Thus, could the authors quantify these gaps with a statistical test (e.g., a paired test over seeds) rather than relying on point estimates, and soften the "on par" language in the abstract/intro to reflect that this holds more precisely on some datasets (Instagram, Reddit) than others (Cora, WikiCS)?
2. **Interpretability evaluation**: The *Necessity and sufficiency* analysis is a concept-count sensitivity. However, it would be interesting to see a per-instance necessity/sufficiency measurement as usually done in the CBM literature. What happens if you delete/insert a concept? Then, the *Sparsity* analysis is based on a case study with only two hand-selected Cora instances. Here, it would be important to add a broader, randomly samples case study across datasets since the human-readability is one of the main selling point of the paper.
3. **Hyperparameter choice**: For instance-based concept extraction, authors mention that $m$ graphs are used. However, I think I am missing the actual value of $m$. Furthermore, the candidate pool size is set to 100 but it is not clear if this parameter is performance sensitive to this choice. Could you please clarify these two points?
4. **Synthetic OOD/adversarial corruptions** The upsampling-ratio-based OOD split and random edge drop/add perturbations are controlled and reproducible, which is a plus for rigour, but they may not fully represent naturally occurring distribution shift (e.g., temporal drift, genuinely novel subpopulations). A brief discussion of how these synthetic settings relate to real-world shift would help readers understand the robustness claims.

**Audience:**

Yes

**Audience Explanation:**

Yes. This paper sits at the intersection of several relevant and active subfields (self-explainable/interpretable GNNs, concept bottleneck models, and LLM-augmented graph learning). In particular, apart from the methodology, the robustness-under-shift analysis and the cross-LLM concept-convergence study can be of independent usefulness: the former offers a concrete, quantified mechanism (spurious-feature suppression via a constrained bottleneck) for why concept-based prediction might generalise better than raw structural reasoning, and the latter addresses a reproducibility concern (LLM-dependence) that affects the growing number of LLM-in-the-loop graph learning papers, not just this one.

**Broader Impact Concerns:**

N/A.

**Claims And Evidence:**

Yes

**Claims Explanation:**

The paper's two main contributions are evaluated with real rigour. The robustness claim ("GCB's advantage over GNNs increases under distribution shifts and adversarial perturbations") is the best-supported claim in the paper. Figures 5 and 6 isolate the shift/perturbation-intensity dependence directly and the pattern (near-zero advantage at IID/clean -> growing advantage under shift) is consistent across four of five datasets, with Cora explicitly and honestly flagged as the exception. The claim that GCB's concept space is not LLM-specific is also well-supported by the four convergence metrics used and the downstream accuracy results reported in Table 4.

The claim that the accuracy is on par with black-box GNNs is weakly supported by the evidence. In fact, while broadly true in aggregate, it is not uniformly supported. It would help adding significance testing to understand if the reported differences are true or not. The interpretability claim is supported by a well-designed experiment (the random-concept substitution reported in Figure 1), but the evidence that the concepts are actually interpretable to a human reader is confined to two hand-selected Cora instances which is illustrative but not a systematic evidence.

**Requested Changes:**

1. **Quantify and soften the "on par with black-box GNNs" claim**: Table 6 shows GCB trailing the best GNN by ~4 F1 points on both Cora and WikiCS in the regular/IID setting. Please add a significance test (e.g., paired test across the 5 seeds) to clarify which gaps are distinguishable from noise, and adjust the abstract/intro phrasing to reflect that parity holds more precisely on some datasets (Instagram, Reddit) than others (Cora, WikiCS).
2. **Strengthen the interpretability evaluation**: The "necessity and sufficiency" analysis is a concept-count sensitivity sweep rather than a per-instance necessity/sufficiency measurement, and the "sparsity"/human-readability claim rests on a two-instance case study (Figure 3). Given that human-readable explanation is GCB's central claim, please add one of: (a) a broader, randomly sampled case study across datasets, (b) a small human evaluation of concept relevance, or (c) a standard comprehensiveness/sufficiency metric (e.g., concept deletion/insertion curves) computed over the full test set.
3. **Report missing concept-retrieval hyperparameters**: Instance-Based Concept Extraction samples $m$ graph instances from each class, but $m$ is not reported. Please also add a brief sensitivity analysis for the candidate pool size to show the performance sensitivity to this specific value.
4. **Discuss the synthetic-vs-real-world nature of the shift settings**: The OOD (label-upsampling) and adversarial (edge-perturbation) settings are controlled and reproducible, but it's unclear how well they approximate naturally occurring distribution shift (e.g., temporal drift, genuinely novel subpopulations). A short discussion would help readers understand the robustness claims.

---
**Minor changes**: please make sure to use parenthetical and textual citations correctly.

---

### Review · Reviewer_t4Gw · 2026-07-20

**Summary Of Contributions:**

Summary

This paper proposes Graph Concept Bottleneck (GCB) for self-explainable node classification on text-attributed graphs. Instead of explaining predictions through salient subgraphs, GCB maps each target node and its local graph into a natural-language concept space and performs classification solely from the resulting concept activations, making concepts part of the actual prediction process.
The framework consists of graph–concept contrastive pretraining, LLM-based candidate concept construction, sparse concept selection, and concept-based classification. The authors evaluate GCB on five datasets under standard, distribution-shift, and structural-perturbation settings. The results show competitive in-distribution performance and improved stability under distribution shifts and random edge perturbations. The paper also analyzes concept-set size, encoder choice, concept leakage, and cross-LLM consistency.

Strengths

1. The core idea is clear, and the explanation format is well suited to text-attributed graphs.
Compared with salient subgraphs, which are often difficult to interpret directly, natural-language concepts better match the semantic nature of text-attributed nodes such as papers, posts, and product descriptions. The paper further integrates these concepts into the prediction pathway rather than generating them only after prediction, strengthening the connection between explanation and model decision-making.

2. The overall framework is complete and practically deployable at inference time.
The graph–concept contrastive pretraining, class-level and instance-level concept generation, sparse concept selection, and concept-based classifier are coherently connected. Large language models are used only during the offline concept construction stage, and no repeated LLM calls are required at inference time, which helps control deployment cost.

3. The experimental coverage is broad, and the paper directly examines concept leakage.
The experiments cover five datasets from different domains, multiple standard and self-explainable graph models, different levels of distribution shift, and various structural perturbation ratios. The random-concept experiment is particularly valuable, as it shows that large concept spaces may still support classification through non-semantic activation patterns, leading to a more cautious and informative discussion of concept faithfulness.

Weaknesses

1. The source of the reported robustness improvements is not sufficiently disentangled.
GCB simultaneously uses external-data pretraining, structural augmentation, a frozen graph encoder, and a concept bottleneck, whereas most baselines do not adopt the same configuration. Therefore, it remains unclear whether the gains under distribution shift and structural perturbation mainly arise from the concept bottleneck itself, or from pretraining, augmentation, or encoder freezing.

2. The theoretical characterization of the “information bottleneck” is somewhat overstated.
The actual training objective mainly consists of a classification loss and an L1 sparsity penalty on the gating vector, without explicitly estimating or optimizing the mutual-information terms in the classical information bottleneck objective. The current module is therefore closer to sparse concept selection than to a strict information bottleneck formulation.

3. The evaluation of explanation faithfulness is still indirect.
The random-concept control can reveal concept leakage, but it does not demonstrate that specific natural-language concepts have a causal effect on individual predictions. In particular, when the concept set is large, random concepts can still achieve relatively strong in-distribution performance, suggesting that the classifier may exploit non-semantic patterns in the activation vector.

4. The experimental setting does not fully support the claim of adversarial robustness.
The structural perturbations consist only of randomly adding and removing edges in the training graph, without optimizing the perturbations against the model, its predictions, or a specific attack objective. The setting is therefore closer to structural noise or random graph corruption than to adaptive graph adversarial attacks.

5. The paper lacks non-LLM baselines for concept construction.
The paper compares concepts generated by multiple large language models and uses random numeric concepts to test leakage, but does not compare against frequency-based terms, TF-IDF keywords, class-discriminative keywords, or conventional keyword extraction methods. It is therefore unclear whether LLMs provide semantic abstraction beyond simple corpus statistics.

**Audience:**

Yes

**Audience Explanation:**

The paper is likely to interest researchers working on graph learning, interpretability, concept bottleneck models, and LLM-assisted representation learning. Its central idea—using natural-language concepts as an intrinsic prediction bottleneck for text-attributed graphs—is relevant and distinct from conventional subgraph-based explanations. The results on distribution shifts, structural perturbations, concept leakage, and cross-LLM consistency also provide useful empirical observations, even though some claims require stronger validation.

**Broader Impact Concerns:**

The work does not introduce an immediate high-risk application.

**Claims And Evidence:**

No

**Claims Explanation:**

The paper provides promising empirical results, but several central claims are not fully supported. The robustness gains are not clearly disentangled from pretraining, structural augmentation, and encoder freezing. Explanation faithfulness is evaluated mainly through indirect random-concept tests rather than direct concept interventions. Moreover, random edge perturbations support robustness to structural noise, but not necessarily to adaptive adversarial attacks. The information-bottleneck claim is also stronger than the implemented L1-regularized gating objective.

**Requested Changes:**

Requested Changes

1. Add matched ablation studies.
The authors should control for pretraining, structural augmentation, and encoder freezing by adding variants without the concept bottleneck, without pretraining, with downstream fine-tuning, and with standard GNNs using the same pretraining configuration. These comparisons are necessary to determine whether the robustness gains are primarily attributable to the concept bottleneck.

2. Reconsider the positioning of the information bottleneck module.
The authors should provide a clearer theoretical connection between the L1-gated objective and the classical information bottleneck formulation. If such a connection cannot be sufficiently justified, the module should be described as “sparse concept selection” or “information-constrained concept optimization,” and claims concerning mutual-information compression should be moderated.

3. Add direct concept intervention experiments.
The authors should test the necessity, sufficiency, and actual decision influence of the selected concepts by removing high-contribution concepts, retaining only key concepts, manually modifying concept activations, and comparing interventions on meaningful and random concepts.

4. Revise the robustness terminology or add genuine adversarial attacks.
If no additional experiments are added, the current setting should be described as “structural perturbation” or “random edge noise.” If the paper retains claims of adversarial robustness, it should include standard graph attacks optimized against the model or specific target nodes.

5. Add non-LLM concept baselines.
The authors should construct concepts using term frequency, TF-IDF, class-discriminative keywords, TextRank, or KeyBERT, and compare them with LLM-generated concepts under the same concept-set size, selection procedure, and classifier configuration. This would directly test whether the LLM-based concept generation module is necessary.

---

### Review · Reviewer_zwFu · 2026-07-21

**Summary Of Contributions:**

This paper proposes Graph Concept Bottleneck (GCB) for self-explainable node classification on text-attributed graphs. GCB aligns graph representations with natural-language concepts through contrastive pretraining, uses LLMs to generate candidate concepts, selects a compact subset, and predicts labels from concept activations. The motivation is clear, the framework is coherent, and the experiments cover five datasets under standard, distribution-shift, and structural-corruption settings.

**Additional Comments:**

The current OOD experiment mainly evaluates changes in class proportions and should be described more specifically as class-distribution or label-prior shift. Random edge addition and deletion should also be described as structural corruption rather than adversarial perturbation unless an optimized attack is used.

**Audience:**

Yes

**Audience Explanation:**

The use of natural-language concept bottlenecks for graph learning is relevant to researchers working on graph representation learning and interpretable machine learning. The paper also raises an interesting question about whether concept-based models genuinely rely on concept semantics or merely use concept activations as latent features.

**Broader Impact Concerns:**

The paper should acknowledge that LLM-generated concepts may contain hallucinations or biases, which is important because these concepts are presented as explanations.

**Claims And Evidence:**

No

**Claims Explanation:**

The predictive results are promising, but several central claims are not yet fully supported.

- The proposed “information bottleneck” is implemented mainly as sparse concept selection using an $L_1$ penalty. The connection to information bottleneck optimization is insufficiently justified, and the claim that the selected concepts are “causally informative” is too strong.

- The source of the robustness improvement is unclear. GCB uses an externally pretrained and frozen graph encoder, with edge perturbations included during pretraining, whereas most baselines are trained from scratch. The reported gains may therefore result from pretraining and augmentation rather than from concept-guided prediction.

- The explanation evaluation lacks direct intervention evidence. For selected classes, removing the concepts considered important for that class and comparing the resulting degradation against random concept removal would provide stronger evidence that these concepts are actually relevant to the predictions.

- The random-concept analysis is useful, although numerical tokens may be a relatively weak control because they are not comparable to normal natural-language concepts.

**Requested Changes:**

- Compare GCB against a baseline using the same pretrained and frozen graph encoder, but directly predicting from the graph representation without the concept bottleneck.
- Provide class-specific concept intervention experiments. For several classes, remove the concepts considered important for that class and compare the performance drop with random concept removal.
- Better justify the information bottleneck formulation, or revise the method description to sparse concept selection and remove unsupported causal claims.
- Replace or complement numerical random concepts with unrelated natural-language concepts or concepts shuffled across classes.